# DNAMotifTokenizer: Towards Biologically Informed Tokenization of Genomic Sequences

## Abstract

DNA language models have advanced genomics, but their downstream performance varies widely due to differences in tokenization, pretraining data, and architecture. We argue that a major bottleneck lies in tokenizing sparse and unevenly distributed DNA sequence motifs, which are critical for accurate and interpretable models. To investigate, we systematically benchmark k-mer and Byte-Pair Encoding (BPE) tokenizers under controlled pretraining budget, evaluating across multiple downstream tasks from five datasets. We find that tokenizer choice induces task-specific trade-offs, and that vocabulary size and tokenizer training data strongly influence the biological knowledge captured. Notably, BPE tokenizers achieve strong performance when trained on smaller but biologically significant data. Building on these insights, we introduce DNAMotifTokenizer, which directly incorporates domain knowledge of DNA sequence motifs into the tokenization process. DNAMotifTokenizer consistently outperforms BPE across diverse benchmarks, demonstrating that knowledge-infused tokenization is crucial for learning powerful, interpretable, and generalizable genomic representations.

## 1 Introduction

Recent advances in artificial intelligence (AI) and large language models (LLMs) have transformed nearly every field of biological research. By analyzing complex, noisy, and large-scale datasets, these models can uncover hidden patterns, generate predictions, and accelerate the discovery of new biological knowledge and molecular structures (Nature Methods, 2024). In genetics, building on the remarkable success of text-based LLMs, researchers have developed DNA language models (DNA-LMs) to capture the latent "grammar" of genomic sequences. These models are being leveraged to improve DNA sequence design, investigate the genetic basis of evolution, and interpret genetic mutations underlying human traits and diseases.

Over the past few years, a series of DNA-LMs has emerged (see Figure 1a). Early efforts, such as DNABERT-1 (Ji et al., 2021), introduced k-mer–based tokenizers and transformer architectures to model DNA sequences, laying the foundation for various downstream applications. DNABERT-2 (Zhou et al., 2023) extended this idea by introducing byte-pair encoding (BPE)-based (Sennrich et al., 2015) tokenizer and pretrained on multi-species genomes. A large-scale model, Nucleotide Transformer (Dalla-Torre et al., 2025), has scaled up in both parameter size and training corpus, improving accuracy and generalizability. The HyenaDNA (Nguyen et al., 2023) has explored long-context modeling to better capture distal dependencies in the genome. More recently, Evo-2 (Brixi et al., 2025) has been developed to expand prediction and design across DNA, RNA, and proteins. Collectively, these models underscore both the promise and challenges of scaling DNA-LMs for biological discovery, including regulatory element prediction, non-coding genetic variant interpretation, and DNA sequence designs.

Despite their superb fine-tuning performance in downstream tasks, current DNA-LMs often exhibit poor zero-shot generalization to new tasks (Patel et al., 2024). The bottleneck lies largely in the DNA tokenization process, which breaks down raw DNA sequences into fundamental units for the model to process (see Figure 1b). Standard tokenization strategies, such as fixed k-mers or subword methods like byte-pair encoding (BPE), often fail to efficiently capture these biologically meaningful DNA motifs. It is critical to optimize the DNA tokenization step towards the development of accurate, interpretable, and generalizable models.

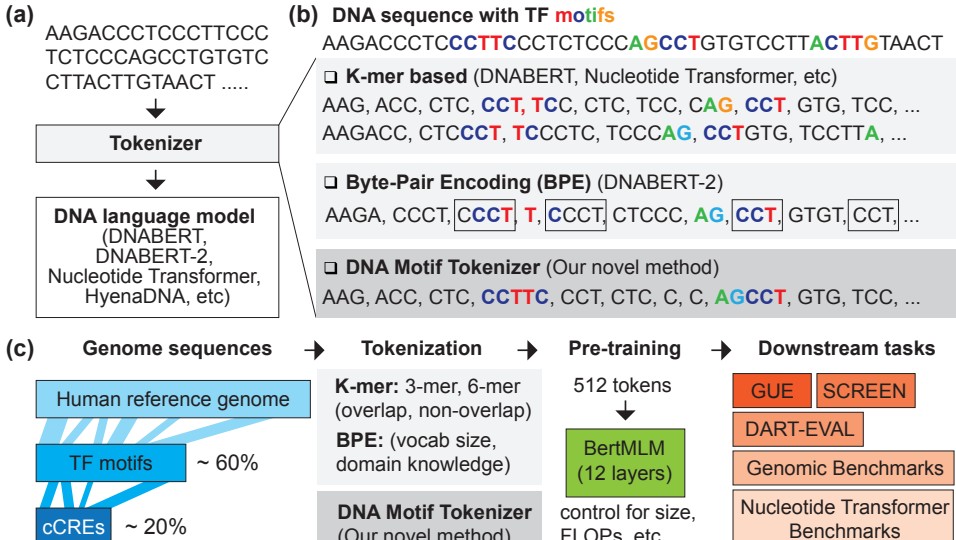

Figure 1: Overview Strategy and Pipeline. **(a)** DNA language-modeling pipeline. **(b)** State-of-the-art genomic tokenizers and DNA Motif Tokenizer. **(c)** Domain knowledge for tokenizer construction, our novel DNAMotifTokenizer, pretraining and downstream evaluation workflow.

In this work, we systematically investigate the impact of tokenization on DNA-LMs with different categories of human genomic sequences (see Figure 1c). Under controlled pretraining settings, we benchmark a variety of k-mer and BPE-based tokenizers across five distinct datasets spanning multiple downstream tasks. Our analysis reveals that the choice of tokenizer induces significant task-specific trade-offs, and we find that incorporating domain knowledge, DNA sequence motifs, is essential for learning more robust DNA representations.

To address this, we introduce DNAMotifTokenizer, a novel strategy that directly incorporates domain knowledge of DNA sequence motifs into the DNA tokenization process. In our comprehensive benchmarks, DNAMotifTokenizer consistently outperforms BPE-based tokenizers across diverse tasks, demonstrating that knowledge-infused tokenization is crucial for learning powerful, interpretable, and generalizable genomic representations.

Our main contributions can therefore be summarized as follows: (1) We introduce Search Candidate cis-Regulatory Elements by ENCODE (SCREEN) benchmarking dataset, which contains a standardized, comprehensive, and well-annotated human functional genomic regulatory elements. (2) We provide the first systematic evaluation of DNA tokenization strategies under controlled pretraining, revealing task-specific trade-offs. (3) We propose DNAMotifTokenizer, a motif-aware tokenizer that directly encodes biologically meaningful sequence motifs. DNAMotifTokenizer consistently outperforms or achieves the state-of-the-art performance, highlighting the necessity of introducing domain knowledge for genomic representation learning. The code, data, and pre-trained model are available in **supplementary materials**.

## 2 BACKGROUND

In natural language processing (NLP), tokenization is a critical step that converts text into a format suitable for computational models. Similarly, genomic sequences can be viewed as a "language" encoding complex information that regulates gene expression in organs, tissues, and cell types across healthy and disease conditions.

The k-mer and BPE-based tokenizers are commonly used in state-of-the-art DNA-LMs(Table 1). HyenaDNA (Nguyen et al., 2023) uses a 1-mer tokenizer and a decoder-only architecture with the Hyena operator to model long-range dependencies. DNABERT-1 (Ji et al., 2021) tokenized the genome with overlapping k-mers and trained separate models for each, substantially outperforming several downstream tasks. Nucleotide Transformer (Dalla-Torre et al., 2025) employs non-overlapping 6-mer tokenization and leverages large-scale pretraining across thousands of human

Table 1: Comparison of state-of-the-art DNA-LMs

| Model | Model Size | Tokenizer | Pretrain Data |
|-------|-----------|-----------|---------------|
| HyenaDNA | $\leq$6.6M | 1mer | Human reference genome |
| DNAbert1-3mer | 86M | 3mer, stride=1 | Human reference genome |
| DNAbert1-6mer | 89M | 6mer, stride=1 | Human reference genome |
| DNAbert2 | 117M | BPE, vocab size=4096 | 135 species genomes |
| NT-HumanRef | 500M | 6mer, stride=6 | Human reference genome |
| NT-1000GG | 500M/2.5B | 6mer, stride=6 | 3202 human genomes |
| NT-HumanRef | 2.5B | 6mer, stride=6 | 850 species genomes |
| MxDNA | 100M | Customized | Human reference genome |

individual genomes, and hundreds of species. Zhou et al. (2023) proposed DNABERT-2, which uses a byte-pair encoding (BPE) tokenizer inspired by NLP. MxDNA (Qiao et al., 2024) introduced a learnable tokenizer using a Mixture-of-Experts framework (Shazeer et al., 2017) and deformable convolutions (Dai et al., 2017).

Despite these advances, the field lacks a systematic comparison of how tokenizer choice alone affects model performance. Existing models differ not only in tokenizers but also in architecture, model size, and pretraining data, which confound direct comparisons (Table 1). This limits our ability to reason about what makes a tokenizer effective and to design better ones.

In addition, DNA is composed of highly repetitive, short, sparse, unevenly distributed, but conserved DNA sequence motifs across 600 million years of bilaterian evolution Nitta et al. (2015), largely represented by transcription factor (TF) binding motifs. The complexity of gene regulation often arises from the specific rearrangement and combination of these conserved motifs into context-specific regulatory elements Wong et al. (2020). Standard tokenization methods, which are agnostic to biological function, often arbitrarily split these meaningful motifs into smaller, non-functional tokens (see Figure 1b). As a result, it hampers DNA-LMs' ability to learn biologically meaningful representations of genomic sequences and complicates downstream model interpretation.

To address these issues, we developed a controlled benchmarking framework to systematically assess the tokenizer's influence and identify key factors for effective design. Guided by our findings, we propose DNAMotifTokenizer, a novel tokenizer that takes a significant step towards a more biologically informed tokenization of genomic sequences. By embedding the essential "grammar" of gene regulation into its vocabulary, DNAMotifTokenizer enables DNA-LMs to better capture the complex relationships between sequence and function. We also assess the generalizability, stability, complexity, and interpretability of our approach.

## 3 DATA

### 3.1 GENOMIC SEQUENCES

**Human reference genome:** We use the most widely used human reference genome, version hg38 (GRCh38) (Consortium, 2013), which incorporates improved accuracy and coverage over previous releases.

**Annotation of motif regions:** We download the genome-wide JASPAR CORE TF motif predictions on the human reference genome (hg38) from the UCSC Genome Browser (Lee et al., 2020)(Raney et al., 2014). For BPE training, we extract all predicted motif sequences and merge any overlapping ones to create a non-redundant set. The resulting set of motif sequences covers approximately 59.84% (See Table C.1) of the human genome.

**Annotation of cCREs:** Candidate cis-regulatory elements (cCREs) are functional regulatory units in the genome, such as promoters and enhancers, that are typically hundreds of base pairs long. These regions often contain multiple TF motifs and are crucial for controlling when and where genes are expressed. For this study, we download human cCRE annotations from The Encyclopedia of DNA Elements(ENCODE), which provide genomic coordinates and regulatory classifications (Moore et al., 2020)(Luo et al., 2020). We extract the corresponding DNA sequences for BPE training, which constitute approximately 20.32% (See Table C.2) of the human reference genome.

## 3.2 BENCHMARK DATASETS

**Genome Understanding Evaluation(GUE):** Genome Understanding Evaluation (GUE) dataset was collected by (Zhou et al., 2023), consisting of 28 distinct datasets across 7 tasks and 4 species, with DNA inputs ranging from 70 to 1000 base pairs (bp). The metric we use is the Matthews Correlation Coefficient (MCC) (see **Appendix**).

**Nucleotide Transformer Benchmarks:** Nucleotide Transformer (NT) benchmarks were collected by (Dalla-Torre et al., 2025), it includes 18 datasets across 4 tasks only on humans. For this dataset, MCC is employed as the metric.

**Dart-Eval:** This dataset was introduced by (Patel et al., 2024) and contains five tasks. In our experiments, we focus on tasks 1–3. Accuracy (ACC) served as the evaluation metric. Task 1 involves distinguishing cCRE regions from background sequences. Task 2 requires identifying transcription factor (TF) binding motifs within background sequences. Task 3 entails classifying sequences specific to five different cell types against background sequences.

**Genomic Benchmarks:** This dataset was collected by (Grešová et al., 2023), it includes 9 tasks, across 9 species. We only use human related datasets. MCC is used as the metric.

**SCREEN:** We create this benchmark dataset by first downloading the cCREs on hg38 from the SCREEN interface (Moore et al., 2020), a platform for searching and visualizing the ENCODE Registry of candidate cis-Regulatory Elements (cCREs). This Registry contains 2,348,854 human cCREs, classified into eight categories, including promoter-like signatures (PLS), proximal and distal enhancer-like signatures (pELS, dELS), and CTCF- or TF–associated accessible elements (CA, CA-CTCF, CA-H3K4me3, CA-TF, TF). We generated a negative superset by taking the complement of all cCRE regions and dividing it into 300 base pair (bp) segments. For each cCRE category, we then randomly sample from this superset to obtain the same number of sequences as in the corresponding negative set. This procedure yielded eight datasets, each containing an equal number of positive and negative sequences.

## 4 METHOD

### 4.1 TOKENIZERS

**K-mer tokenizer:** A k-mer is a substring of length k from a DNA or RNA sequence (Moeckel et al., 2024). K-mers can be generated in two ways: overlapping, by sliding a window one base at a time to capture all subsequences, or non-overlapping, by moving the window in steps of k to produce disjoint subsequences. In our experiments, we test overlapping 3-mer and 6-mer implemented by DNABERT-1 (Ji et al., 2021), and non-overlapping 6-mer implemented by Nucleotide Transformer (Dalla-Torre et al., 2025), respectively.

**BPE tokenizer:** BPE is a learnable subword tokenization algorithm that iteratively merges the most frequent pairs of characters or subwords in a training corpus to build a vocabulary of common subwords (Sennrich et al., 2015). The resulting merge rules, based on pair frequency, are recorded and applied to tokenize new sequences consistently, allowing the model to capture recurring patterns efficiently. The BPE tokenizer for DNA sequences was introduced by DNABERT-2. We use the BPE tokenizer pretrained by DNABERT-2 and also train our own BPEs on human reference genome, motif enriched genomic regions, and cis-regulatory element (cCRE) regions (see **Section 3**). Three vocabulary sizes were explored: 4096 (matching DNABERT-2), 2048, and 1024. Each BPE was initialized with the DNA alphabet (A, T, C, G, N) and five special tokens ([PAD], [UNK], [CLS], [SEP], [MASK]), with a minimum frequency threshold of 100.

**Others:** MxDNA (Qiao et al., 2024) is excluded from comparison due to a lack of publicly available pre-trained weights.

**DNAMotifTokenizer:** We develop DNAMotifTokenizer, a novel tokenizer designed to understand the language of the genome by directly embedding biological domain knowledge—specifically, TF motifs, the functional 'words' of the genome—into its vocabulary. In addition, we design a greedy algorithm to tokenize the incoming sequences with our customized vocabulary. See **Section 7** for more details.

## 4.2 MODELS

**Architecture:** We adopt a BERT-based masked language model architecture (BertMLM) (Devlin et al., 2019) for our experiments. The model uses the Transformer encoder architecture (Vaswani et al., 2017) with 12 layers and 12 attention heads and a hidden size of 768, resulting in intermediate feed-forward layers of size 3072. The maximum input sequence length is 512 tokens, and the model uses learnable positional embeddings up to this length. Dropout Srivastava et al. (2014) is applied to both the attention probabilities and hidden layers with a rate of 0.1, and the GELU (Hendrycks & Gimpel, 2016) activation function is used throughout. The model vocabulary size is changed according to the tokenizer used, and a type vocabulary of size 2 is included to distinguish segment embeddings. All parameters are initialized with a standard deviation of 0.02.

**Pre-training:** During pre-training, we strictly control all experiments to ensure comparability by keeping the models' floating-point operations per second (FLOPs) consistent. All tokenizers are applied to the human reference genome (chromosomes 1–22, X, Y, and M), and the tokenized sequences are sequentially split into segments of 512 tokens to serve as input for the BertMLMs. Segments containing more than 50% N tokens are discarded. Each model is trained with a batch size of 96 for 200,000 steps, using a learning rate of 4e-5, Adam optimizer parameters ($\beta_1 = 0.9, \beta_2 = 0.98, \delta = 1 \times 10^{-6}$), weight decay of 0.01, a masked language modeling probability of 0.15, and 10,000 warmup steps.

**Evaluation:** During model evaluation, we fine-tune the pre-trained models on five benchmark datasets (see Section 3 for details). Most fine-tuning hyperparameters are consistent across models, varying primarily in the maximum input length, which is adjusted per tokenizer. Performances are measured using the Matthews Correlation Coefficient (MCC) for four datasets, and Accuracy (ACC) for the DART-Eval benchmarking dataset.

We compare the vocabulary similarities by calculating the pairwise Jaccard Index. We also generate Venn diagrams illustrating their word overlap (Hulsen, 2021). See **Appendix** for more details.

## 5 BENCHMARKING OF STATE-OF-THE-ART TOKENIZERS

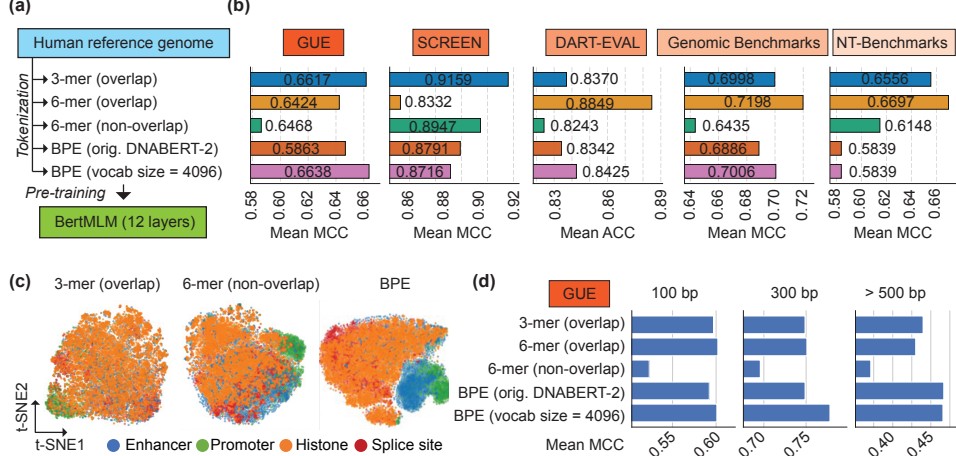

Figure 2: Benchmarking of state-of-the-art Tokenizers. **(a)** Overview benchmarking pipeline for overlap k-mer, non-overlap k-mer, and BPE-based tokenizers trained by DNABERT-2 and trained on human genome (vocabulary size 4,096). **(b)** Evaluation on five benchmarking datasets. **(c)** Zero-shot performance of k-mer and BPE-based tokenizer on genomic region classification. **(d)** BPE-based tokenizer outperforms in longer DNA sequence from GUE dataset.

A variety of k-mer and BPE-based tokenizers are used in state-of-the-art DNA-LMs. However, these models vary in architecture, model size, and pre-training data (see Table 1), making it difficult to isolate the impact of the tokenizer on model performance. To systematically evaluate various DNA tokenizers, we design experiments on the human reference genome with strictly controlled pre-training and fine-tuning protocols (see **Section 4.2**). These DNA tokenizers include 3-mer (overlap), 6-mer (overlap) from DNABERT-1, 6-mer (non-overlap) from Nucleotide Transformer, the original BPE from DNABERT-2, and a custom BPE tokenizer (vocabulary size = 4096) trained on the human ref-

erence genome (see Figure 2a). We fine-tune each pre-trained model on five distinct benchmarking datasets and compare the average performance of each DNA tokenizer (Figure 2b). Our findings reveal a clear performance trade-off: although k-mer-based tokenizers achieve higher performance on specific tasks, like splicing site prediction, no single tokenizer consistently outperformed the others. Overall, BPE tokenizers achieve more robust performance, surpassing k-mer tokenizers in four of the five benchmark datasets. More detailed results for each task across all benchmark datasets are provided in the **Appendix**.

We use the NT-benchmarks to investigate the zero-shot ability, where k-mer tokenizers consistently outperformed BPE tokenizers. Both overlapping and non-overlapping k-mer tokenizers struggled to cluster four distinct genomic elements, while BPE tokenizers performed better at separating enhancers and promoters from other DNA sequences (Figure 2c). Furthermore, we leverage the most comprehensive GUE benchmark and group downstream tasks by the length of DNA sequences. We find that k-mer tokenizers performed marginally better on shorter sequences (100 bp), whereas BPE tokenizers excel on longer ones (>500 bp) (Figure 2d), which is consistent with previous work (Zhou et al., 2023). Taken together, we conclude that BPE-based tokenizers achieve more robust performance on both zero-shot and fine-tuning tasks and are more generalizable for predictions involving longer DNA sequences.

## 6    WHAT MAKES A GOOD BPE TOKENIZER?

To better optimize BPE tokenizers for genomic sequences, we examine two key dimensions: vocabulary size, which determines token granularity and model performance, and domain knowledge, which grounds tokens in biologically meaningful units.

### 6.1    SIZE MATTERS: VOCABULARY SCALING

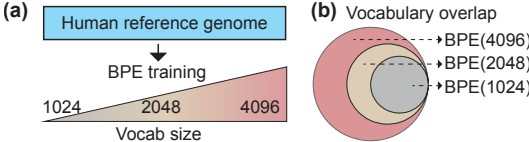

Figure 3: BPE with Different Vocabulary Size. **(a)** Training BPEs with vocabularies of 1,024, 2,048, and 4,096 tokens. **(b)** Vocabulary overlap across the trained BPEs.

Using the human reference genome, we first train BPE tokenizers with three vocabulary sizes: 1024, 2048, and 4096 (Figure 3a). We overlap token vocabularies and find that they were nested (Figure 3b). For example, each larger set (e.g. BPE 2048) contains all tokens from the smaller ones (e.g., BPE 1024). This hierarchical structure means longer vocabularies are built by merging shorter, existing tokens, leading to a marginal increase in average token length (Figure B.1a) and a wider distribution of token frequencies (Figure B.1 b). Next, to quantify the information gained by expanding the vocabulary, we compute the Shannon entropy for the set of novel tokens introduced at each increase in vocabulary size. For instance, BPE (2048-1024) refers to the new tokens learned by the BPE 2048-token vocabulary that were not present in the BPE 1024-token vocabulary, with the same logic applying to BPE (4096-2048). We observe a significant increase (Wilcoxon test) in mean entropy (red triangle) across BPE (1024), BPE (2048-1024) and BPE (4096-2048), suggesting that larger vocabularies capture more complex and diverse tokens within the genomic sequences (Figure B.1c). Last, we fine-tune models using each of these BPE tokenizers on five benchmark datasets to evaluate their downstream performance (Table 2). Counterintuitively, the more diverse and longer tokens captured by a larger vocabulary did not translate to better predictive performance. In fact, our findings consistently show that a more concise token vocabulary led to superior overall results.

In summary, we suggest that increasing the BPE vocabulary size beyond a certain point introduces information redundancy, which may negatively impact model performance.

### 6.2    INFORMATIVE TOKENS: ADDING DOMAIN KNOWLEDGE

We hypothesize that training BPE tokenizers on biologically meaningful subsets of the genome, rather than the entire genome, could yield more effective models. To test our hypothesis, we train

Table 2: Performance of BPE models with varying vocabulary sizes across five benchmark datasets

| Model | GUE Ave. MCC | SCREEN Ave. MCC | DART-EVAL Ave. ACC | Genomic Benchmarks Ave. MCC | NT-Benchmarks Ave. MCC |
|---|---|---|---|---|---|
| BPE(DNABERT2) | 0.6468 ±0.1242 | **0.8791** ±0.0224 | 0.8342 ±0.0328 | 0.6886 ±0.1721 | 0.5839 ±0.1016 |
| BPE(4096) | 0.6638 ±0.122 | 0.8717 ±0.0294 | 0.8425 ±0.0408 | 0.7006 ±0.1565 | 0.5839 ±0.0945 |
| BPE(2048) | 0.6684 ±0.1302 | 0.8743 ±0.033 | 0.8451 ±0.0531 | 0.7021 ±0.1584 | 0.595 ±0.1052 |
| BPE(1024) | **0.673** ±0.127 | 0.8781 ±0.0269 | **0.8604** ±0.0471 | **0.7069** ±0.1556 | **0.5988** ±0.1121 |

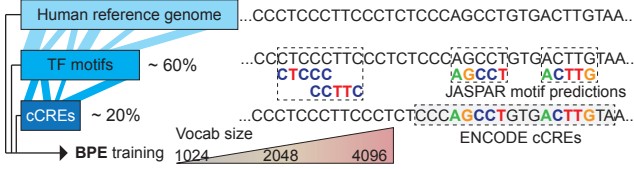

Figure 4: Training BPE on Domain Knowledge. **Left:** Train BPE tokenizers on the whole human genome, TF motif regions (60% of the genome), and cCRE regions (20% of the genome) with different vocabulary size. **Right:** Illustration of TF motifs and cCREs distributions on the genome.

BPE tokenizers with various vocabulary sizes on three distinct datasets, including the full human reference genome, genomic regions predicted as TF motifs, and biological function-enriched cCREs regions, to assess whether data selection improves tokenization (see **Section 3** and Figure 4). We first compare the BPE token vocabularies learned from the human reference genome, motif, and cCRE regions by calculating their pairwise Jaccard Similarity Index (Figure B.2a). Our analysis reveals that the BPE tokenizer trained on motif regions produced the most distinct vocabulary, compared to those trained on the whole genome or cCREs. For example, with a vocabulary size of 1024, the motif-trained BPE learned 415 unique tokens (40.5%) not found in the other two vocabularies (Figure B.2b). Although we find no significant differences in token length distribution (Figure B.2c), the motif BPE generated a higher proportion of low-frequency tokens when applied to the human reference genome (Figure B.2d). We also observe the same pattern for BPE tokenizers with larger vocabulary sizes. Next, we rigorously evaluate and compare the downstream performance of models trained with each tokenizer (Tabel 3). Our results show that models with BPE tokenizers trained on motif and cCRE regions can achieve comparable performance to those trained on the whole human genome, regardless of vocabulary size.

These results, consistent across five benchmark datasets, suggest that BPE tokenizers can be trained more efficiently using curated datasets enriched with domain knowledge, without sacrificing model performance.

Table 3: Performance of BPE models with varying vocabulary size and domain knowledge

| Vocab Size | Domain Knowledge | GUE Ave. MCC | SCREEN Ave. MCC | DART-EVAL Ave. ACC | Genomic Benchmarks Ave. MCC | NT-Benchmarks Ave. MCC |
|---|---|---|---|---|---|---|
| 4096 | hg38 | **0.6638** ±0.122 | 0.8717 ±0.0294 | **0.8425** ±0.0408 | **0.7006** ±0.1565 | **0.5839** ±0.0945 |
| | cCREs | 0.6552 ±0.1268 | 0.8719 ±0.0333 | **0.8425** ±0.0451 | 0.699 ±0.1642 | 0.5723 ±0.0896 |
| | motifs | 0.6522 ±0.1299 | 0.8694 ±0.0338 | 0.8309 ±0.0331 | 0.6844 ±0.1644 | 0.5764 ±0.0997 |
| 2048 | hg38 | 0.6684 ±0.1302 | 0.8743 ±0.033 | **0.8451** ±0.0531 | **0.7021** ±0.1584 | 0.595 ±0.1052 |
| | cCREs | 0.6645 ±0.1283 | 0.8767 ±0.0279 | 0.8382 ±0.0416 | 0.6993 ±0.1685 | **0.5973** ±0.1108 |
| | motifs | **0.6695** ±0.1273 | **0.8769** ±0.0231 | 0.8392 ±0.0499 | 0.6868 ±0.1617 | 0.5902 ±0.1057 |
| 1024 | hg38 | 0.673 ±0.127 | 0.8781 ±0.0269 | **0.8604** ±0.0471 | **0.7069** ±0.1556 | **0.5988** ±0.1121 |
| | cCREs | **0.6743** ±0.1244 | **0.8793** ±0.0242 | 0.8458 ±0.0556 | 0.7039 ±0.1633 | 0.5954 ±0.1097 |
| | motifs | 0.6684 ±0.1278 | 0.8777 ±0.0244 | 0.8494 ±0.0478 | 0.7005 ±0.163 | 0.5954 ±0.1143 |

# 7 OUR NOVEL DNAMOTIFTOKENIZER

Building on aforementioned insights, we would like to ask whether domain knowledge can be directly integrated into the tokenizer's design to improve performance. To test it, we develop DNAMotifTokenizer, a novel tokenizer that directly incorporates TF motifs into the vocabulary and applies a greedy search algorithm to tokenize the corresponding patterns in DNA sequences (Figure 5).

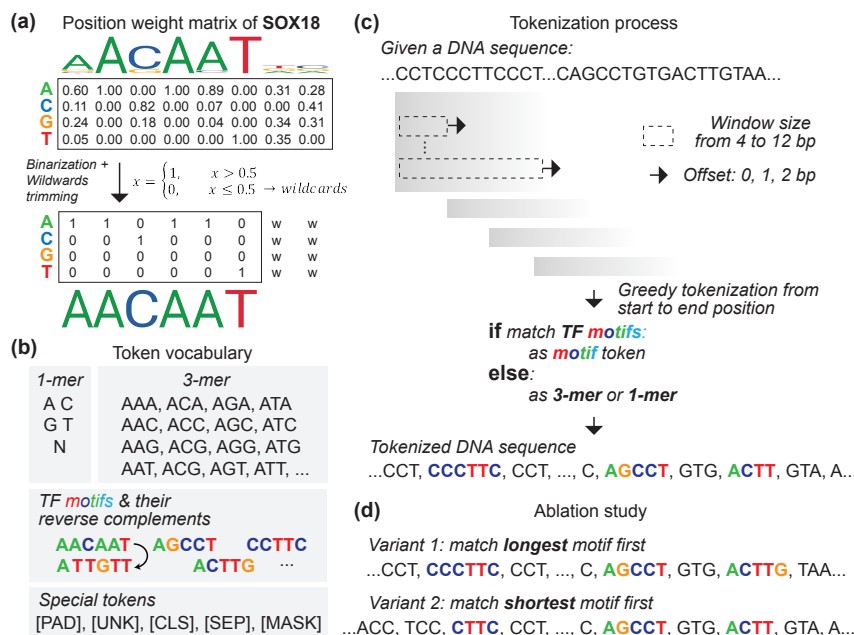

Figure 5: Overview of DNAMotifTokenizer. **(a)** Pre-processing of motif's probability weight matrix. **(b)** The constructed vocabulary, consisting of special tokens, motif tokens, 3-mer tokens, and 1-mer tokens. **(c)** The greedy tokenization algorithm, which by default randomly selects among matched motifs at each position. **(d)** Two deterministic variants of the tokenizer: matching by the longest motif first or by the shortest motif first.

## 7.1 ALGORITHM

**Motif processing:** Motifs are short, recurring patterns in DNA, RNA, or protein sequences that are frequently associated with specific biological functions. In this work, we focus on TF motifs, which are short DNA sequences that are bound by transcription factor proteins to regulate gene expression. The TF motifs are generally represented in the form of position weight matrices (PWM)(Stormo, 2000), which indicate the probability of each nucleotide occurring at each position within the motif. We download motif PWMs from the JASPAR 2024 motif library (Sandelin et al., 2004)(Rauluse-viciute et al., 2024), which is a widely used, open-access repository. We use the vertebrate library, which contains 879 non-redundant motifs. As most TF motifs range in length from 5 to 12 base pairs (bp), we exclude any motifs longer than 12 bp from our analysis. To incorporate TF motifs into our vocabulary, we binarize their PWMs and encode them into fixed sequences (Figure 5a). For each TF PWM, we apply a threshold probability of 0.5. Positions with lower probability are defined as wildcard positions. Subsequently, we discard wildcard positions at both ends of the motif. For the remaining positions, we encode them using the nucleotide with the highest probability.

**Our customized token vocabulary:** In addition to the motif sequences defined in the previous section, we include their reverse complements in the vocabulary. Furthermore, we add 3-mer, 1-mer(A, T, C, G, N), and five special tokens, namely `[PAD]`, `[UNK]`, `[CLS]`, `[SEP]`, and `[MASK]`, to form our final vocabulary. Our final vocabulary has 901 tokens in total, including 827 motif sequences, 64 3-mer, 5 1-mer(A, T, C, G, N), and 5 special tokens (Figure 5b).

**Tokenization algorithm:** With our customized vocabulary established, we implement a greedy, non-overlapping tokenization algorithm. The algorithm scans an incoming DNA sequence from left to right, using a sliding window that varies from 4 to 12 bp in length. We recognize that a single base-pair shift can cascade and alter the entire tokenized output. To mitigate this sensitivity without a significant loss in computational efficiency, we incorporate local flexibility by allowing for a 0 to 2 bp offset to the right. At each position, if the subsequence within the window matches one or more motifs in our vocabulary, one is selected at random to serve as the token. If no motif match is found, the sequence is tokenized using fallback 3-mer or 1-mer representations (Figure 5c). The pseudocode is shown as Algorithm 1 in **Appendix**.

## 7.2 EVALUATION PERFORMANCE

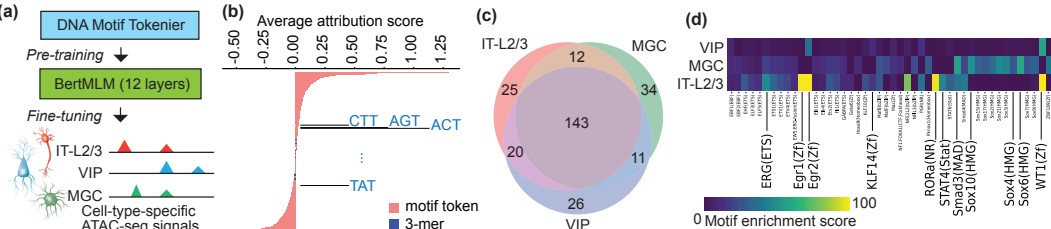

Figure 6: Interpretation of DNAMotifTokenizer. **(a)** Workflow for the multiclass cell-type ATAC-seq classification task. **(b)** Distribution of average attribution scores for k-mer and motif tokens on the test set (10%). The motif tokens exhibit the strongest influence on model predictions. **(c)** Overlap between top 200 highest-attribution tokens from three cell types. **(d)** Highly attributed motif tokens matched with motif enrichment results from Li et al. (2023).

**Interpretability:** We collected single-nucleus ATAC-seq (snATAC-seq) data generated from three diverse brain cell types: intratelencephalic neurons from cortical layer 2/3 (ITL23), VIP-positive GABAergic neurons (VIP), and Microglia (MGC) Li et al. (2023). We identified cell-type-specific ATAC-seq peak regions, which serve as prediction targets (Figure 6a). To control the vocabulary size and achieve fair comparison, we then fine-tuned our DNAMotifTokenizer models on this dataset. Then, we performed token attribution analysis using Integrated Gradients Sundararajan et al. (2017) on the test set (10%). For each token, we computed its attribution score toward the true class label and then averaged the scores; Tokens with attribution scores farther away from zero—either positive or negative—indicate stronger influence on the model's prediction (Figure 6b). By ranking them, we found that in our DNAMotifTokenizer model, most of the highly influential tokens correspond to motif tokens (in red), whereas the k-mer tokens (in blue) generally had attribution scores close to zero. We selected the top 200 most contributive motif tokens for each cell type. The Venn plot indicates that DNAMotifTokenizer not only utilizes 143 motif tokens shared between three cell types, but also uses 25, 26, 34 cell-type-specific motif tokens for prediction in ITL23, VIP, MGC, respectively (Figure 6c). In addition, we compared these most contributive motif tokens with the enriched motifs for each cell type reported in the original paper Li et al. (2023). We showed top-matched motifs from three cell types as a heatmap (Figure 6d). In ITL23 cell type, motifs from the WT1, RORa, Egr, and KLF families are captured and enriched, whereas only WT1, and Egr2 are captured in the VIP cell type. In MGC cell type, SOX family motifs are captured and enriched. These results demonstrate the interpretability of motif tokens introduced by DNAMotifTokenizer in various cell types.

**Downstream tasks:** We evaluate the downstream performance of models trained with DNAMotifTokenizer. We reveal that DNAMotifTokenizer consistently matches or exceeds the performance of BPE models (Table 4). For DART-EVAL tasks, the synthetic sequences may not exhibit a natural motif distribution, which can affect the classification performance of our method.

**Generalizability:** We finetuned our models on Yeast and Mouse datasets from the GUE dataset. Compared to published k-mer and BPE tokenizers learnt from multiple species, DNAMotifTokenizer demonstrates comparable or superior performance in these cross-species predictions. Specifically, DNAMotifTokenizer achieves average MCC at 0.4662 in yeast, 0.5509 in mouse (Table F.6). These results confirm the robustness and generalizability of DNAMotifTokenizer even outside human datasets.

**Stability:** We use four different seeds to run the tokenization algorithm and train four different models, and finetuned on the benchmark datasets, computed the mean and standard variation across each subdataset within each benchmark datasets. (Table G.1 G.2 G.3 G.4 G.5).

## 7.3 COMPLEXITY ANALYSIS

**Time Complexity:** The average token length in our vocabulary is approximately 8.3 nucleotides. For a genomic sequence of length $n$, the tokenizer proceeds sequentially from left to right, resulting in approximately $O\left(\frac{n}{8.3}\right)$ tokenization steps. At each position, the tokenizer queries a Trie to identify motif tokens with lengths ranging from 4 up to Maxlen. If no motif is found, the tokenizer

Table 4: Performance of DNAMotifTokenizer and all BPEs, on five Benchmark datasets

| Model | GUE Ave. MCC | SCREEN Ave. MCC | DART-EVAL Ave. ACC | Genomic Benchmark Ave. MCC | NT-Benchmarks Ave. MCC |
|---|---|---|---|---|---|
| BPE(DNABERT2) | $0.6468 \pm_{0.1242}$ | $0.8791 \pm_{0.0224}$ | $0.8342 \pm_{0.0328}$ | $0.6886 \pm_{0.1721}$ | $0.5839 \pm_{0.1016}$ |
| BPE(hg38, 4096) | $0.6638 \pm_{0.122}$ | $0.8717 \pm_{0.0294}$ | $0.8425 \pm_{0.0408}$ | $0.7006 \pm_{0.1565}$ | $0.5839 \pm_{0.0945}$ |
| BPE(hg38, 2048) | $0.6684 \pm_{0.1302}$ | $0.8743 \pm_{0.033}$ | $0.8451 \pm_{0.0531}$ | $0.7021 \pm_{0.1584}$ | $0.595 \pm_{0.1052}$ |
| BPE(hg38, 1024) | $0.673 \pm_{0.127}$ | $0.8781 \pm_{0.0269}$ | $\mathbf{0.8604} \pm_{0.0471}$ | $\mathbf{0.7069} \pm_{0.1556}$ | $0.5988 \pm_{0.1121}$ |
| BPE(motifs, 4096) | $0.6522 \pm_{0.1299}$ | $0.8694 \pm_{0.0338}$ | $0.8309 \pm_{0.0331}$ | $0.6844 \pm_{0.1644}$ | $0.5764 \pm_{0.0997}$ |
| BPE(motifs, 2048) | $0.6695 \pm_{0.1273}$ | $0.8769 \pm_{0.0231}$ | $0.8392 \pm_{0.0499}$ | $0.6868 \pm_{0.1617}$ | $0.5902 \pm_{0.1057}$ |
| BPE(motifs, 1024) | $0.6684 \pm_{0.1278}$ | $0.8777 \pm_{0.0244}$ | $\underline{0.8494} \pm_{0.0478}$ | $0.7005 \pm_{0.163}$ | $0.5954 \pm_{0.1143}$ |
| BPE(cCREs, 4096) | $0.6552 \pm_{0.1268}$ | $0.8719 \pm_{0.0333}$ | $0.8425 \pm_{0.0451}$ | $0.699 \pm_{0.1642}$ | $0.5723 \pm_{0.0896}$ |
| BPE(cCREs, 2048) | $0.6645 \pm_{0.1283}$ | $0.8767 \pm_{0.0279}$ | $0.8382 \pm_{0.0416}$ | $0.6993 \pm_{0.1685}$ | $0.5973 \pm_{0.1108}$ |
| BPE(cCREs, 1024) | $0.6743 \pm_{0.1244}$ | $0.8793 \pm_{0.0242}$ | $0.8458 \pm_{0.0556}$ | $\underline{0.7039} \pm_{0.1633}$ | $0.5954 \pm_{0.1097}$ |
| DNAMotifTokenizer (default) | $\mathbf{0.6815} \pm_{0.1236}$ | $\mathbf{0.885} \pm_{0.0217}$ | $0.8437 \pm_{0.0574}$ | $0.6976 \pm_{0.1522}$ | $\mathbf{0.6018} \pm_{0.1167}$ |
| **Ablation** | | | | | |
| DNAMotifTokenizer (longest) | $0.6687 \pm_{0.1311}$ | $\underline{0.8822} \pm_{0.0237}$ | $0.8388 \pm_{0.0567}$ | $0.6884 \pm_{0.1639}$ | $0.5965 \pm_{0.1072}$ |
| DNAMotifTokenizer (shortest) | $\underline{0.6697} \pm_{0.1238}$ | $0.8809 \pm_{0.0303}$ | $0.8399 \pm_{0.0617}$ | $0.6884 \pm_{0.1549}$ | $\underline{0.6003} \pm_{0.1107}$ |

defaults to 3-mer or 1-mer tokens. The worst-case Trie lookup cost is $O(\text{Maxlen}^2)$. Considering potential shifts by 1 or 2 nucleotides at each position, the upper bound of the total time complexity is $O\left(\frac{n}{8.3} \cdot \text{Maxlen}^2\right)$. In our implementation, we set $\text{Maxlen} = 12$.

**Space Complexity:** Let $V$ denote the vocabulary size and $L$ the average token length. The Trie storing all motif tokens requires $O(V \cdot L)$ space, while the lookup table mapping motifs to token IDs takes at most $O(V)$. During tokenization, the output tokens occupy $O\left(\frac{n}{8.3}\right)$ space. Therefore, the total space complexity is $O\left(\frac{n}{8.3} + V \cdot L\right)$. In our implementation, $V = 901$, the average token length is $L \approx 8.3$, and the number of motif tokens stored in the Trie is 827.

## 7.4 ABLATION STUDY

To evaluate the motif selection strategy, we conduct an ablation study. We test two deterministic variants: one that greedily selects the longest possible motif at each position, and another that selects the shortest (Figure 5d). While these greedy variants were still highly effective, outperforming all BPE models on three of five benchmarks, neither of these greedy strategies outperforms our default algorithm (Table 4). This finding is consistent with our earlier results, suggesting that simply optimizing for token length, and/or the nucleotide diversity in tokens, does not necessarily improve model performance. This underscores the complexity of genomic "grammar" and highlights the potential for developing more sophisticated motif selection strategies in future work.

## 8 CONCLUSION

In this work, we first introduce the SCREEN benchmark, a comprehensive dataset of well-annotated human functional genomic regulatory elements. Together with other benchmark datasets, we systematically evaluated state-of-the-art k-mer and BPE tokenizers under controlled settings, revealing a clear performance trade-off across different downstream tasks.

We investigated BPE optimization, finding that simply increasing vocabulary size can introduce information redundancy that harms model performance. Instead, we demonstrate that BPE tokenizers can be trained more efficiently on smaller, curated DNA sequences enriched with domain knowledge. Building on these insights, we introduce DNAMotifTokenizer, a novel tokenizer whose performance is SOTA. While our algorithm is suboptimal due to manual injections in tokenization, we highlight the necessity of incorporating domain knowledge for genomic representation learning.

Our future research will focus on (1) enhancing our approach by improving the flexibility of motif representations within the vocabulary, (2) developing a learnable tokenization algorithm, and (3) further exploring the downstream benefits for model interpretability. The ultimate goal is to advance the development of biologically informed tokenization and interpretation of genomic sequences.

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

# APPENDIX

## A DNAMOTIFTOKENIZER: TOKENIZATION ALGORITHM

---

**Algorithm 1:** Tokenization algorithm for DNAMotifTokenizer

---

**Data:** Sequence $s$ with length $n$, Vocabulary $V = \{\text{motifs, 3-mer, 1-mer}\}$
**Result:** $y = [tokens]$
**Function** Tokenize($i$)
    $score, best\_token, candidates \leftarrow -1, \text{None}, [\,]$
    **for** $j \leftarrow 4$ **to** 12 **do**
      $seg \leftarrow$ substring of $s$ from index $i$ to $i + j$
      **if** $seg \in motifs$ **then**
        Append $seg$ to $candidates$;

    **if** $candidates \neq None$ **then**
      $best\_token \leftarrow$ random element from $candidates$;
    **else**
      **if** $i + 3 > n$ **then**
        $best\_token \leftarrow$ 1-mer at $s[i]$;
      **else**
        $best\_token \leftarrow$ 3-mer at $s[i : i + 3]$;

    $next\_pos \leftarrow i+$ length of $best\_token$
    $best\_score \leftarrow$ length of $best\_token$
    **return** $best\_token, best\_score, next\_pos$;
**Function** Main()
    $i, y \leftarrow 0, [\,]$
    **while** $i < n$ **do**
      $best\_token, best\_score, next\_pos \leftarrow \text{None}, \text{-1}, \text{i}$
      **for** $offset \leftarrow 0$ **to** 2 **do**
        $candidate\_token, candidate\_score, candidate\_pos \leftarrow$ Tokenize($i + offset$)
        **if** $candidate\_score > best\_score$ **then**
          $best\_token \leftarrow candidate\_token$;
          $best\_score \leftarrow candidate\_score$;
          $next\_pos \leftarrow candidate\_pos$;

      $i \leftarrow next\_pos$
      Append $best\_token$ to $y$;
    **return** $y$;

---

# B WHAT MAKES A GOOD BPE TOKENIZER?

## B.1 SIZE MATTERS: VOCABULARY SCALING

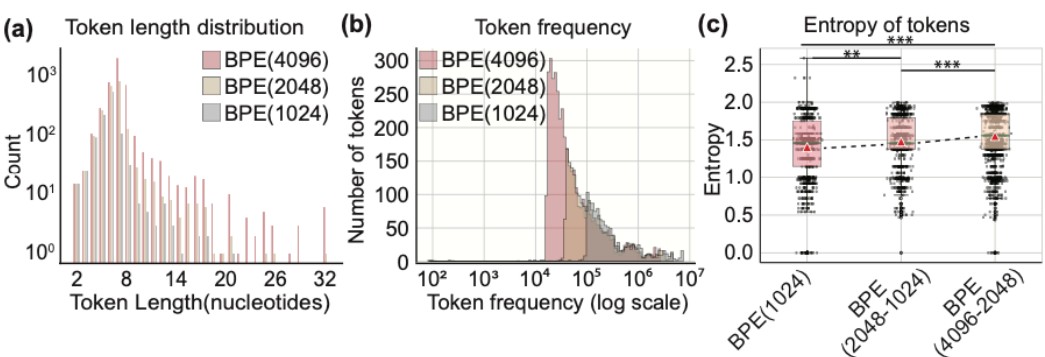

Figure B.1: Comparison across BPEs with Different Vocabulary Size. Wilcoxon test, ***, *p*-value < 0.001, **, *p*-value < 0.01

## B.2 INFORMATIVE TOKENS: ADDING DOMAIN KNOWLEDGE

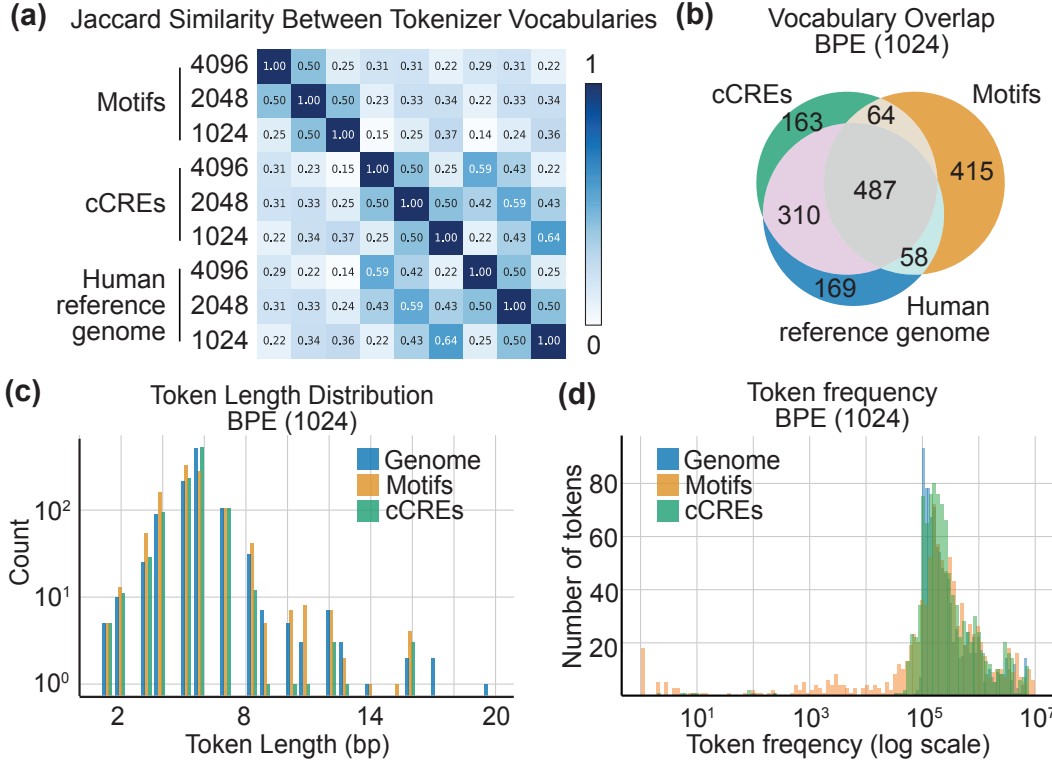

Figure B.2: Comparison across BPEs with Different Vocabulary Size and Domain Knowledge.

## C  MOTIF AND CCRE REGIONS

Table C.1: Jaspar Motif Annotation on hg38 Genome

| Genome | Total Nucleotides at Motif Regions | Ratio (%) |
|---|---|---|
| hg38 | 1,848,048,414 | 59.84 |

Table C.2: cCRE Regions in hg38 Genome

| Genome | Total cCRE Regions | Total Nucleotides at cCRE Regions | Ratio (%) |
|---|---|---|---|
| hg38 | 2,348,854 | 627,448,729 | 20.32 |

## D  BENCHMARKING OF STATE-OF-THE-ART TOKENIZERS

Table D.1: Performance comparison of models across datasets on GUE, grouped by task.

| Dataset | 3mer (overlap) | 6mer (overlap) | 6mer (non-overlap) | BPE (orig. DNABERT-2) | BPE (vocab size=4096) |
|---|---|---|---|---|---|
| prom_core_all | 0.6645 | 0.6497 | 0.5842 | 0.5966 | 0.6270 |
| prom_core_notata | 0.6745 | 0.6783 | 0.6233 | 0.6421 | 0.6435 |
| prom_core_tata | 0.5640 | 0.4748 | 0.4919 | 0.5938 | 0.6417 |
| prom_300_all | 0.8730 | 0.8336 | 0.7871 | 0.8240 | 0.8304 |
| prom_300_notata | 0.9089 | 0.9022 | 0.8631 | 0.8926 | 0.9037 |
| prom_300_tata | 0.4643 | 0.5170 | 0.4372 | 0.5297 | 0.6006 |
| reconstructed | 0.8064 | 0.6434 | 0.6328 | 0.7217 | 0.7143 |
| tf_0 | 0.6506 | 0.6537 | 0.6151 | 0.6402 | 0.6560 |
| tf_1 | 0.7078 | 0.6908 | 0.6326 | 0.6611 | 0.6952 |
| tf_2 | 0.5184 | 0.5357 | 0.4680 | 0.5587 | 0.5603 |
| tf_3 | 0.4496 | 0.4304 | 0.3164 | 0.4349 | 0.4234 |
| tf_4 | 0.6588 | 0.6993 | 0.5842 | 0.6657 | 0.6692 |
| **Mean** | **0.6617** | **0.6424** | **0.5863** | **0.6468** | **0.6638** |
| Std | 0.1487 | 0.1386 | 0.1482 | 0.1242 | 0.122 |

Table D.2: Performance comparison of models across datasets on Genomic Benchmarks

| Dataset | 3mer (overlap) | 6mer (overlap) | 6mer (non-overlap) | BPE (orig. DNABERT-2) | BPE (vocab size=4096) |
|---|---|---|---|---|---|
| human_enhancers_ensembl | 0.7514 | 0.7771 | 0.6310 | 0.7182 | 0.7438 |
| human_nontata_promoters | 0.8116 | 0.8228 | 0.6975 | 0.8081 | 0.7937 |
| demo_coding_vs_intergenomic_seqs | 0.8225 | 0.8340 | 0.7720 | 0.8113 | 0.8173 |
| human_enhancers_cohn | 0.4890 | 0.4824 | 0.4360 | 0.4565 | 0.4820 |
| human_ensembl_regulatory | 0.8240 | 0.8294 | 0.8678 | 0.8466 | 0.8413 |
| human_ocr_ensembl | 0.5005 | 0.5734 | 0.4569 | 0.4909 | 0.5254 |
| **Mean** | **0.6998** | **0.7199** | **0.6435** | **0.6886** | **0.7006** |
| Std | 0.1611 | 0.1528 | 0.1719 | 0.1721 | 0.1565 |

Table D.3: Performance comparison of models across datasets on Nucleotide transformer benchmarks

| Dataset | 3mer (overlap) | 6mer (overlap) | 6mer (non-overlap) | BPE (orig. DNABERT-2) | BPE (vocab size=4096) |
|---|---|---|---|---|---|
| H2AFZ | 0.4727 | 0.5077 | 0.4758 | 0.4757 | 0.4916 |
| H3K27ac | 0.4460 | 0.4728 | 0.4080 | 0.4688 | 0.4914 |
| H3K27me3 | 0.5720 | 0.5781 | 0.6077 | 0.5861 | 0.5983 |
| H3K36me3 | 0.5972 | 0.6066 | 0.5814 | 0.6061 | 0.5955 |
| H3K4me1 | 0.4598 | 0.4662 | 0.4629 | 0.4793 | 0.4804 |
| H3K4me2 | 0.5692 | 0.5867 | 0.5665 | 0.5790 | 0.5741 |
| H3K4me3 | 0.6537 | 0.6672 | 0.6486 | 0.6622 | 0.6736 |
| H3K9ac | 0.5201 | 0.5435 | 0.4985 | 0.5316 | 0.5390 |
| H3K9me3 | 0.4227 | 0.4276 | 0.3741 | 0.4215 | 0.4271 |
| H4K20me1 | 0.6064 | 0.6114 | 0.6068 | 0.6195 | 0.6074 |
| splice_sites_donors | 0.9776 | 0.9679 | 0.9503 | 0.6612 | 0.6557 |
| splice_sites_acceptors | 0.9721 | 0.9577 | 0.9163 | 0.6865 | 0.6297 |
| splice_sites_all | 0.9737 | 0.9655 | 0.9046 | 0.6507 | 0.5845 |
| enhancers_types | 0.5270 | 0.5596 | 0.4629 | 0.4636 | 0.4697 |
| enhancers | 0.5799 | 0.6123 | 0.4888 | 0.4741 | 0.5090 |
| promoter_no_tata | 0.7800 | 0.8169 | 0.7483 | 0.7450 | 0.7429 |
| promoter_tata | 0.8792 | 0.9093 | 0.6412 | 0.6629 | 0.7097 |
| promoter_all | 0.7909 | 0.7984 | 0.7228 | 0.7356 | 0.7299 |
| **Mean** | **0.6556** | **0.6697** | **0.6147** | **0.5839** | **0.5839** |
| Std | 0.1909 | 0.1838 | 0.1745 | 0.1016 | 0.0945 |

Table D.4: Performance comparison of models across datasets on SCREEN

| Dataset | 3mer (overlap) | 6mer (overlap) | 6mer (non-overlap) | BPE (orig. DNABERT-2) | BPE (vocab size=4096) |
|---|---|---|---|---|---|
| CA-CTCF | 0.9099 | 0.9126 | 0.8880 | 0.8705 | 0.8702 |
| pELS | 0.9179 | 0.9182 | 0.9008 | 0.8897 | 0.8869 |
| CA | 0.9179 | 0.9196 | 0.8875 | 0.8777 | 0.8778 |
| CA-H3K4me3 | 0.9132 | 0.9146 | 0.8850 | 0.8743 | 0.8657 |
| CA-TF | 0.9116 | 0.9126 | 0.8836 | 0.8357 | 0.8085 |
| TF | 0.9089 | 0.9086 | 0.8881 | 0.8747 | 0.8629 |
| PLS | 0.9342 | 0.9352 | 0.9240 | 0.9098 | 0.9013 |
| dELS | 0.9138 | 0.2440 | 0.9003 | 0.9006 | 0.9000 |
| **Mean** | **0.9159** | **0.8332** | **0.8947** | **0.8791** | **0.8717** |
| Std | 0.0081 | 0.2382 | 0.0135 | 0.0224 | 0.0294 |

Table D.5: Performance comparison across tasks on Dart-Eval(task1-3)

| Dataset | 3mer (overlap) | 6mer (overlap) | 6mer (non-overlap) | BPE (orig. DNABERT-2) | BPE (vocab size=4096) |
|---|---|---|---|---|---|
| task1 | 0.8668 | 0.8734 | 0.7976 | 0.8618 | 0.8631 |
| task2 | 0.9919 | 0.9924 | 0.9599 | 0.8923 | 0.9327 |
| GM12878 | 0.7914 | 0.8684 | 0.7896 | 0.8235 | 0.8227 |
| H1ESC | 0.7775 | 0.8758 | 0.7968 | 0.8178 | 0.8266 |
| HEPG2 | 0.8165 | 0.8639 | 0.8225 | 0.8334 | 0.8312 |
| IMR90 | 0.7612 | 0.8716 | 0.7619 | 0.7942 | 0.7982 |
| K562 | 0.8534 | 0.8490 | 0.8419 | 0.8164 | 0.8232 |
| **Mean** | **0.8370** | **0.8849** | **0.8243** | **0.8342** | **0.8425** |
| Std | 0.0847 | 0.0465 | 0.0620 | 0.0328 | 0.0447 |

# E    WHAT MAKES A GOOD BPE TOKENIZER?

Table E.1: Performance comparison across datasets on GUE, with BPE trained on cCREs, varying in vocabulary size.

| Dataset | BPE (cCREs, 4096) | BPE (cCREs, 2048) | BPE (cCREs, 1024) |
|---|---|---|---|
| prom_core_all | 0.6173 | 0.6224 | 0.6102 |
| prom_core_notata | 0.6363 | 0.6405 | 0.6458 |
| prom_core_tata | 0.6040 | 0.7018 | 0.7130 |
| prom_300_all | 0.8338 | 0.8308 | 0.8459 |
| prom_300_notata | 0.9073 | 0.9069 | 0.9017 |
| prom_300_tata | 0.5921 | 0.6074 | 0.5965 |
| reconstructed | 0.6903 | 0.7383 | 0.7660 |
| tf_0 | 0.6522 | 0.6431 | 0.6773 |
| tf_1 | 0.6722 | 0.6662 | 0.6583 |
| tf_2 | 0.5361 | 0.5437 | 0.5637 |
| tf_3 | 0.4200 | 0.4052 | 0.4384 |
| tf_4 | 0.7007 | 0.6673 | 0.6750 |
| **Mean** | **0.6552** | **0.6645** | **0.6743** |
| Std | 0.1268 | 0.1283 | 0.1244 |

Table E.2: Performance comparison across datasets on GUE, with BPE trained on motifs, varying in vocabulary size.

| Dataset | BPE (motifs, 4096) | BPE (motifs, 2048) | BPE (motifs, 1024) |
|---|---|---|---|
| prom_core_all | 0.6200 | 0.6236 | 0.6250 |
| prom_core_notata | 0.6296 | 0.6352 | 0.6338 |
| prom_core_tata | 0.5838 | 0.6811 | 0.6847 |
| prom_300_all | 0.8274 | 0.8255 | 0.8294 |
| prom_300_notata | 0.8941 | 0.9005 | 0.9014 |
| prom_300_tata | 0.5796 | 0.6310 | 0.5767 |
| reconstructed | 0.7362 | 0.7810 | 0.7694 |
| tf_0 | 0.6574 | 0.6368 | 0.6385 |
| tf_1 | 0.6820 | 0.6915 | 0.7056 |
| tf_2 | 0.5508 | 0.5425 | 0.5336 |
| tf_3 | 0.3947 | 0.4133 | 0.4309 |
| tf_4 | 0.6708 | 0.6716 | 0.6913 |
| **Mean** | **0.6522** | **0.6695** | **0.6684** |
| Std | 0.1299 | 0.1273 | 0.1278 |

Table E.3: Performance comparison across datasets on GUE, with BPE trained on hg38, varying in vocabulary size.

| Dataset | BPE (hg38, 4096) | BPE (hg38, 2048) | BPE (hg38, 1024) |
|---|---|---|---|
| prom_core_all | 0.6270 | 0.6349 | 0.6254 |
| prom_core_notata | 0.6435 | 0.6503 | 0.6571 |
| prom_core_tata | 0.6417 | 0.6482 | 0.6842 |
| prom_300_all | 0.8304 | 0.8388 | 0.8298 |
| prom_300_notata | 0.9037 | 0.9039 | 0.9028 |
| prom_300_tata | 0.6006 | 0.5483 | 0.5799 |
| reconstructed | 0.7143 | 0.7867 | 0.7763 |
| tf_0 | 0.6560 | 0.6655 | 0.6714 |
| tf_1 | 0.6952 | 0.6759 | 0.6999 |
| tf_2 | 0.5603 | 0.5392 | 0.5370 |
| tf_3 | 0.4234 | 0.4353 | 0.4309 |
| tf_4 | 0.6692 | 0.6940 | 0.6818 |
| **Mean** | **0.6638** | **0.6684** | **0.6730** |
| Std | 0.122 | 0.1302 | 0.127 |

Table E.4: Performance comparison across datasets on Nucleotide Transformer Benchmarks, with BPE trained on cCREs, varying in vocabulary size.

| Dataset | BPE (cCREs, 4096) | BPE (cCREs, 2048) | BPE (cCREs, 1024) |
|---|---|---|---|
| H2AFZ | 0.4809 | 0.4682 | 0.4751 |
| H3K27ac | 0.4765 | 0.4759 | 0.4657 |
| H3K27me3 | 0.6047 | 0.6108 | 0.6052 |
| H3K36me3 | 0.5775 | 0.5950 | 0.5950 |
| H3K4me1 | 0.4889 | 0.4801 | 0.4909 |
| H3K4me2 | 0.5642 | 0.5712 | 0.5917 |
| H3K4me3 | 0.6524 | 0.6686 | 0.6812 |
| H3K9ac | 0.5308 | 0.5317 | 0.5447 |
| H3K9me3 | 0.4192 | 0.4328 | 0.4309 |
| H4K20me1 | 0.6171 | 0.6201 | 0.6282 |
| splice_sites_donors | 0.5983 | 0.7676 | 0.7837 |
| splice_sites_acceptors | 0.6461 | 0.7101 | 0.7047 |
| splice_sites_all | 0.5647 | 0.6909 | 0.6526 |
| enhancers_types | 0.4754 | 0.4624 | 0.4595 |
| enhancers | 0.4950 | 0.4999 | 0.4871 |
| promoter_no_tata | 0.7424 | 0.7406 | 0.7451 |
| promoter_tata | 0.6514 | 0.6932 | 0.6438 |
| promoter_all | 0.7161 | 0.7314 | 0.7313 |
| **Mean** | **0.5723** | **0.5973** | **0.5954** |
| Std | 0.0896 | 0.1108 | 0.1097 |

Table E.5: Performance comparison across datasets on Nucleotide Transformer Benchmarks, with BPE trained on motifs, varying in vocabulary size.

| Dataset | BPE (motifs, 4096) | BPE (motifs, 2048) | BPE (motifs, 1024) |
|---|---|---|---|
| H2AFZ | 0.4762 | 0.4797 | 0.4685 |
| H3K27ac | 0.4628 | 0.4600 | 0.4666 |
| H3K27me3 | 0.5967 | 0.6054 | 0.5977 |
| H3K36me3 | 0.5829 | 0.5859 | 0.5852 |
| H3K4me1 | 0.4803 | 0.4898 | 0.4810 |
| H3K4me2 | 0.5578 | 0.5613 | 0.5788 |
| H3K4me3 | 0.6752 | 0.6734 | 0.6725 |
| H3K9ac | 0.5273 | 0.5294 | 0.5347 |
| H3K9me3 | 0.3995 | 0.4153 | 0.4089 |
| H4K20me1 | 0.6238 | 0.6380 | 0.6172 |
| splice_sites_donors | 0.6544 | 0.7533 | 0.7596 |
| splice_sites_acceptors | 0.5967 | 0.6624 | 0.7157 |
| splice_sites_all | 0.6109 | 0.6435 | 0.6757 |
| enhancers_types | 0.4682 | 0.4712 | 0.4565 |
| enhancers | 0.4974 | 0.5086 | 0.5079 |
| promoter_no_tata | 0.7427 | 0.7434 | 0.7529 |
| promoter_tata | 0.6932 | 0.6670 | 0.6969 |
| promoter_all | 0.7291 | 0.7365 | 0.7408 |
| **Mean** | **0.5764** | **0.5902** | **0.5954** |
| Std | 0.0997 | 0.1057 | 0.1143 |

Table E.6: Performance comparison across datasets on Nucleotide Transformer Benchmarks, with BPE trained on hg38, varying in vocabulary size.

| Dataset | BPE (hg38, 4096) | BPE (hg38, 2048) | BPE (hg38, 1024) |
|---|---|---|---|
| H2AFZ | 0.4916 | 0.4711 | 0.4810 |
| H3K27ac | 0.4914 | 0.4916 | 0.4802 |
| H3K27me3 | 0.5983 | 0.6100 | 0.6074 |
| H3K36me3 | 0.5955 | 0.6013 | 0.5950 |
| H3K4me1 | 0.4804 | 0.4956 | 0.4899 |
| H3K4me2 | 0.5741 | 0.5718 | 0.5835 |
| H3K4me3 | 0.6736 | 0.6844 | 0.6927 |
| H3K9ac | 0.5390 | 0.5332 | 0.5497 |
| H3K9me3 | 0.4271 | 0.4039 | 0.4073 |
| H4K20me1 | 0.6074 | 0.6275 | 0.6214 |
| splice_sites_donors | 0.6557 | 0.7315 | 0.7448 |
| splice_sites_acceptors | 0.6297 | 0.6726 | 0.7071 |
| splice_sites_all | 0.5845 | 0.6406 | 0.6963 |
| enhancers_types | 0.4697 | 0.4914 | 0.4573 |
| enhancers | 0.5090 | 0.4962 | 0.4909 |
| promoter_no_tata | 0.7429 | 0.7509 | 0.7605 |
| promoter_tata | 0.7097 | 0.7120 | 0.6779 |
| promoter_all | 0.7299 | 0.7251 | 0.7350 |
| **Mean** | **0.5839** | **0.5950** | **0.5988** |
| Std | 0.0945 | 0.1052 | 0.1121 |

Table E.7: Performance comparison across datasets on Genomic Benchmarks, with BPE trained on cCREs, varying in vocabulary size.

| Dataset | BPE (cCREs, 4096) | BPE (cCREs, 2048) | BPE (cCREs, 1024) |
|---|---|---|---|
| human_enhancers_ensembl | 0.7386 | 0.7371 | 0.7473 |
| human_nontata_promoters | 0.8035 | 0.8144 | 0.8038 |
| demo_coding_vs_intergenomic_seqs | 0.8224 | 0.8206 | 0.8208 |
| human_enhancers_cohn | 0.4716 | 0.4602 | 0.4681 |
| human_ensembl_regulatory | 0.8442 | 0.8474 | 0.8528 |
| human_ocr_ensembl | 0.5136 | 0.5164 | 0.5307 |
| **Mean** | **0.6990** | **0.6993** | **0.7039** |
| Std | 0.1642 | 0.1685 | 0.1633 |

Table E.8: Performance comparison across datasets on Genomic Benchmarks, with BPE trained on motifs, varying in vocabulary size.

| Dataset | BPE (motifs, 4096) | BPE (motifs, 2048) | BPE (motifs, 1024) |
|---|---|---|---|
| human_enhancers_ensembl | 0.7250 | 0.7325 | 0.7428 |
| human_nontata_promoters | 0.7795 | 0.7657 | 0.7984 |
| demo_coding_vs_intergenomic_seqs | 0.8081 | 0.8078 | 0.8230 |
| human_enhancers_cohn | 0.4632 | 0.4568 | 0.4631 |
| human_ensembl_regulatory | 0.8377 | 0.8445 | 0.8458 |
| human_ocr_ensembl | 0.4927 | 0.5137 | 0.5301 |
| **Mean** | **0.6844** | **0.6868** | **0.7005** |
| Std | 0.1644 | 0.1617 | 0.163 |

Table E.9: Performance comparison across datasets on Genomic Benchmarks, with BPE trained on hg38, varying in vocabulary size.

| Dataset | BPE (hg38, 4096) | BPE (hg38, 2048) | BPE (hg38, 1024) |
|---|---|---|---|
| human_enhancers_ensembl | 0.7438 | 0.7449 | 0.7440 |
| human_nontata_promoters | 0.7937 | 0.7873 | 0.7997 |
| demo_coding_vs_intergenomic_seqs | 0.8173 | 0.8290 | 0.8246 |
| human_enhancers_cohn | 0.4820 | 0.4740 | 0.4810 |
| human_ensembl_regulatory | 0.8413 | 0.8435 | 0.8477 |
| human_ocr_ensembl | 0.5254 | 0.5340 | 0.5445 |
| **Mean** | **0.7006** | **0.7021** | **0.7069** |
| Std | 0.1565 | 0.1584 | 0.1556 |

Table E.10: Performance comparison across datasets on Dart-Eval (task1-3), with BPE trained on cCREs, varying in vocabulary size.

| Dataset | BPE (cCREs, 4096) | BPE (cCREs, 2048) | BPE (cCREs, 1024) |
|---|---|---|---|
| task1 | 0.8629 | 0.8663 | 0.8722 |
| task2 | 0.9430 | 0.9270 | 0.9704 |
| GM12878 | 0.8228 | 0.8165 | 0.8146 |
| H1ESC | 0.8262 | 0.8257 | 0.8223 |
| HEPG2 | 0.8332 | 0.8298 | 0.8289 |
| IMR90 | 0.7972 | 0.7988 | 0.7913 |
| K562 | 0.8120 | 0.8034 | 0.8210 |
| **Mean** | **0.8425** | **0.8382** | **0.8458** |
| Std | 0.0451 | 0.0431 | 0.0602 |

Table E.11: Performance comparison across datasets on Dart-Eval (task1-3), with BPE trained on motifs, varying in vocabulary size.

| Dataset | BPE (motifs, 4096) | BPE (motifs, 2048) | BPE (motifs, 1024) |
|---|---|---|---|
| task1 | 0.8599 | 0.8652 | 0.8675 |
| task2 | 0.8971 | 0.9494 | 0.9594 |
| GM12878 | 0.8192 | 0.8210 | 0.8245 |
| H1ESC | 0.8177 | 0.8197 | 0.8218 |
| HEPG2 | 0.8223 | 0.8262 | 0.8356 |
| IMR90 | 0.7913 | 0.7909 | 0.8118 |
| K562 | 0.8088 | 0.8018 | 0.8248 |
| **Mean** | **0.8310** | **0.8392** | **0.8493** |
| Std | 0.0331 | 0.0499 | 0.0478 |

Table E.12: Performance comparison across datasets on Dart-Eval (task1-3), with BPE trained on hg38, varying in vocabulary size.

| Dataset | BPE (hg38, 4096) | BPE (hg38, 2048) | BPE (hg38, 1024) |
|---|---|---|---|
| task1 | 0.8631 | 0.8656 | 0.8677 |
| task2 | 0.9327 | 0.9658 | 0.9722 |
| GM12878 | 0.8227 | 0.8216 | 0.8415 |
| H1ESC | 0.8266 | 0.8229 | 0.8380 |
| HEPG2 | 0.8312 | 0.8331 | 0.8408 |
| IMR90 | 0.7982 | 0.7977 | 0.8276 |
| K562 | 0.8232 | 0.8093 | 0.8351 |
| **Mean** | **0.8425** | **0.8451** | **0.8604** |
| Std | 0.0408 | 0.0531 | 0.0471 |

Table E.13: Performance comparison across datasets on SCREEN, with BPE trained on cCREs, varying in vocabulary size.

| Dataset | BPE (cCREs, 4096) | BPE (cCREs, 2048) | BPE (cCREs, 1024) |
|---|---|---|---|
| CA-CTCF | 0.8723 | 0.8760 | 0.8785 |
| pELS | 0.8898 | 0.8924 | 0.8953 |
| CA | 0.8774 | 0.8817 | 0.8865 |
| CA-H3K4me3 | 0.8669 | 0.8710 | 0.8734 |
| CA-TF | 0.7974 | 0.8145 | 0.8245 |
| TF | 0.8675 | 0.8738 | 0.8793 |
| PLS | 0.9032 | 0.9036 | 0.8962 |
| dELS | 0.9006 | 0.9003 | 0.9008 |
| **Mean** | **0.8719** | **0.8767** | **0.8793** |
| Std | 0.0333 | 0.0279 | 0.0242 |

Table E.14: Performance comparison across datasets on SCREEN, with BPE trained on motifs, varying in vocabulary size.

| Dataset | BPE (motifs, 4096) | BPE (motifs, 2048) | BPE (motifs, 1024) |
|---|---|---|---|
| CA-CTCF | 0.8719 | 0.8726 | 0.8770 |
| pELS | 0.8873 | 0.8910 | 0.8930 |
| CA | 0.8754 | 0.8814 | 0.8840 |
| CA-H3K4me3 | 0.8620 | 0.8687 | 0.8715 |
| CA-TF | 0.7940 | 0.8295 | 0.8234 |
| TF | 0.8646 | 0.8707 | 0.8751 |
| PLS | 0.9006 | 0.9016 | 0.8962 |
| dELS | 0.8990 | 0.8997 | 0.9013 |
| **Mean** | **0.8694** | **0.8769** | **0.8777** |
| Std | 0.0338 | 0.0231j | 0.0244 |

Table E.15: Performance comparison across datasets on SCREEN, with BPE trained on hg38, varying in vocabulary size.

| Dataset | BPE (hg38, 4096) | BPE (hg38, 2048) | BPE (hg38, 1024) |
|---|---|---|---|
| CA-CTCF | 0.8702 | 0.8752 | 0.8775 |
| pELS | 0.8869 | 0.8916 | 0.8927 |
| CA | 0.8778 | 0.8820 | 0.8843 |
| CA-H3K4me3 | 0.8657 | 0.8678 | 0.8745 |
| CA-TF | 0.8085 | 0.7998 | 0.8173 |
| TF | 0.8629 | 0.8722 | 0.8752 |
| PLS | 0.9013 | 0.9052 | 0.9022 |
| dELS | 0.9000 | 0.9007 | 0.9011 |
| **Mean** | **0.8717** | **0.8743** | **0.8781** |
| Std | 0.0294 | 0.033 | 0.0269 |

# F DNAMOTIFTOKENIZER

Table F.1: Performance comparison across datasets on GUE, with DNAMotifTokenizer, varying in motif matching.

| Dataset | DNAMotifTokenizer (longest) | DNAMotifTokenizer (shortest) | DNAMotifTokenizer |
|---|---|---|---|
| prom_core_all | 0.6412 | 0.6522 | 0.6488 |
| prom_core_notata | 0.6599 | 0.6413 | 0.6564 |
| prom_core_tata | 0.7230 | 0.7228 | 0.7195 |
| prom_300_all | 0.8355 | 0.8389 | 0.8538 |
| prom_300_notata | 0.9014 | 0.8945 | 0.9062 |
| prom_300_tata | 0.6147 | 0.5950 | 0.6441 |
| reconstructed | 0.7544 | 0.7603 | 0.7693 |
| tf_0 | 0.6283 | 0.6379 | 0.6406 |
| tf_1 | 0.6813 | 0.6581 | 0.6670 |
| tf_2 | 0.4907 | 0.4883 | 0.5179 |
| tf_3 | 0.4249 | 0.4756 | 0.4651 |
| tf_4 | 0.6686 | 0.6711 | 0.6888 |
| **Mean** | **0.6687** | **0.6697** | **0.6815** |
| Std | 0.1483 | 0.1348 | 0.1367 |

Table F.2: Performance comparison across datasets on SCREEN, with DNAMotifTokenizer, varying in motif matching.

| Dataset | DNAMotifTokenizer (longest) | DNAMotifTokenizer (shortest) | DNAMotifTokenizer |
|---|---|---|---|
| CA-CTCF | 0.8781 | 0.8808 | 0.8822 |
| pELS | 0.8977 | 0.8972 | 0.8961 |
| CA | 0.8889 | 0.8890 | 0.8888 |
| CA-H3K4me3 | 0.8764 | 0.8782 | 0.8795 |
| CA-TF | 0.8288 | 0.8105 | 0.8380 |
| TF | 0.8861 | 0.8820 | 0.8826 |
| PLS | 0.8982 | 0.9045 | 0.9087 |
| dELS | 0.9038 | 0.9048 | 0.9039 |
| **Mean** | **0.8822** | **0.8809** | **0.8850** |
| Std | 0.0245 | 0.0324 | 0.0253 |

Table F.3: Performance comparison across datasets on Nucleotide Transformer Benchmarks, with DNAMotifTokenizer, varying in motif matching.

| Dataset | DNAMotifTokenizer (longest) | DNAMotifTokenizer (shortest) | DNAMotifTokenizer |
|---|---|---|---|
| H2AFZ | 0.4873 | 0.4848 | 0.4835 |
| H3K27ac | 0.4822 | 0.4735 | 0.4892 |
| H3K27me3 | 0.6043 | 0.6019 | 0.5994 |
| H3K36me3 | 0.5963 | 0.5932 | 0.5915 |
| H3K4me1 | 0.4834 | 0.4930 | 0.4814 |
| H3K4me2 | 0.5890 | 0.5843 | 0.5775 |
| H3K4me3 | 0.6757 | 0.6729 | 0.6701 |
| H3K9ac | 0.5264 | 0.5381 | 0.5434 |
| H3K9me3 | 0.4177 | 0.4205 | 0.4170 |
| H4K20me1 | 0.6197 | 0.6158 | 0.6324 |
| splice_sites_donors | 0.7230 | 0.7346 | 0.7855 |
| splice_sites_acceptors | 0.7172 | 0.7263 | 0.7309 |
| splice_sites_all | 0.6661 | 0.7079 | 0.7119 |
| enhancers_types | 0.4639 | 0.4672 | 0.4507 |
| enhancers | 0.5054 | 0.5053 | 0.4902 |
| promoter_no_tata | 0.7465 | 0.7544 | 0.7577 |
| promoter_tata | 0.6935 | 0.6788 | 0.6855 |
| promoter_all | 0.7396 | 0.7523 | 0.7342 |
| **Mean** | **0.5965** | **0.6003** | **0.6018** |
| Std | 0.1115 | 0.1123 | 0.1154 |

Table F.4: Performance comparison across datasets on Genomic Benchmarks, with DNAMotifTokenizer, varying in motif matching.

| Dataset | DNAMotifTokenizer (longest) | DNAMotifTokenizer (shortest) | DNAMotifTokenizer |
|---|---|---|---|
| human_enhancers_ensembl | 0.7364 | 0.7184 | 0.7426 |
| human_nontata_promoters | 0.7797 | 0.7606 | 0.7717 |
| demo_coding_vs_intergenomic_seqs | 0.8207 | 0.8256 | 0.8248 |
| human_enhancers_cohn | 0.4366 | 0.4408 | 0.4505 |
| human_ensembl_regulatory | 0.8651 | 0.8622 | 0.8629 |
| human_ocr_ensembl | 0.4922 | 0.5227 | 0.5332 |
| **Mean** | **0.6884** | **0.6884** | **0.6976** |
| Std | 0.1748 | 0.1720 | 0.1581 |

Table F.5: Performance comparison across datasets on Dart-Eval (task1-3), with DNAMotifTokenizer, varying in motif matching.

| Dataset | DNAMotifTokenizer (longest) | DNAMotifTokenizer (shortest) | DNAMotifTokenizer |
|---|---|---|---|
| task1 | 0.8609 | 0.8639 | 0.8647 |
| task2 | 0.9525 | 0.9610 | 0.9571 |
| GM12878 | 0.8098 | 0.8105 | 0.8206 |
| H1ESC | 0.8177 | 0.8155 | 0.8202 |
| HEPG2 | 0.8271 | 0.8286 | 0.8292 |
| IMR90 | 0.7862 | 0.7902 | 0.7926 |
| K562 | 0.8172 | 0.8095 | 0.8217 |
| **Mean** | **0.8414** | **0.8414** | **0.8433** |
| Std | 0.0588 | 0.0613 | 0.0586 |

Table F.6: Performance of different models on Species Yeast and Mouse from GUE

| Model | Epigenetic Marks Prediction(Yeast) | Transcription Factor Prediction(Mouse) |
|---|---|---|
| 3mer(stride=1) | 0.4399 | 0.4034 |
| 6mer(stride=1) | 0.4301 | 0.4466 |
| 6mer(stride=6) | 0.3711 | 0.3056 |
| BPE(DNABERT-2) | **0.4670** | **0.5509** |
| DNAMotifTokenizer (longest) | 0.4639 | 0.5306 |
| DNAMotifTokenizer (shortest) | 0.4609 | 0.5458 |
| DNAMotifTokenizer | **0.4662** | **0.5509** |

## G  STABILITY

Table G.1: Performance of DNAMotifTokenizer on GUE, across 4 seeds

| Dataset | Seed 1 | Seed 2 | Seed 3 | Seed 4 | Mean | Std |
|---|---|---|---|---|---|---|
| prom_core_all | 0.6488 | 0.6362 | 0.6484 | 0.6483 | 0.6454 | 0.0063 |
| prom_core_notata | 0.6564 | 0.6530 | 0.6496 | 0.6613 | 0.6551 | 0.0051 |
| prom_core_tata | 0.7195 | 0.7164 | 0.7164 | 0.6999 | 0.7131 | 0.0091 |
| prom_300_all | 0.8538 | 0.8412 | 0.8307 | 0.8351 | 0.8402 | 0.0095 |
| prom_300_notata | 0.9062 | 0.8967 | 0.9003 | 0.8994 | 0.9007 | 0.0040 |
| prom_300_tata | 0.6441 | 0.6049 | 0.6147 | 0.5625 | 0.6066 | 0.0332 |
| reconstructed | 0.7693 | 0.7620 | 0.7232 | 0.7474 | 0.7505 | 0.0202 |
| tf_0 | 0.6406 | 0.6326 | 0.6631 | 0.6277 | 0.6410 | 0.0153 |
| tf_1 | 0.6670 | 0.6985 | 0.7011 | 0.6871 | 0.6884 | 0.0152 |
| tf_2 | 0.5179 | 0.5715 | 0.5037 | 0.5000 | 0.5233 | 0.0327 |
| tf_3 | 0.4651 | 0.4742 | 0.4401 | 0.4431 | 0.4556 | 0.0162 |
| tf_4 | 0.6888 | 0.7039 | 0.6725 | 0.6778 | 0.6858 | 0.0138 |

Table G.2: Performance of DNAMotifTokenizer on Nucleotide Transformer Benchmarks, across 4 seeds

| Dataset | Seed 1 | Seed 2 | Seed 3 | Seed 4 | Mean | Std |
|---|---|---|---|---|---|---|
| H2AFZ | 0.4835 | 0.5106 | 0.4831 | 0.4888 | 0.4915 | 0.0122 |
| H3K27ac | 0.4892 | 0.4663 | 0.4579 | 0.4850 | 0.4746 | 0.0142 |
| H3K27me3 | 0.5994 | 0.6002 | 0.5970 | 0.6014 | 0.5995 | 0.0018 |
| H3K36me3 | 0.5915 | 0.5777 | 0.5852 | 0.5957 | 0.5875 | 0.0074 |
| H3K4me1 | 0.4814 | 0.4762 | 0.4763 | 0.4944 | 0.4821 | 0.0083 |
| H3K4me2 | 0.5775 | 0.5938 | 0.5771 | 0.5816 | 0.5825 | 0.0076 |
| H3K4me3 | 0.6701 | 0.6705 | 0.6612 | 0.6748 | 0.6692 | 0.0058 |
| H3K9ac | 0.5434 | 0.5437 | 0.5426 | 0.5452 | 0.5437 | 0.0011 |
| H3K9me3 | 0.4170 | 0.4218 | 0.4202 | 0.4279 | 0.4217 | 0.0044 |
| H4K20me1 | 0.6324 | 0.6277 | 0.6179 | 0.6183 | 0.6241 | 0.0069 |
| splice sites donors | 0.7855 | 0.7881 | 0.7994 | 0.7304 | 0.7759 | 0.0306 |
| splice sites acceptors | 0.7309 | 0.7288 | 0.7653 | 0.7350 | 0.7400 | 0.0165 |
| splice sites all | 0.7119 | 0.7072 | 0.7598 | 0.6592 | 0.7095 | 0.0418 |
| enhancers types | 0.4507 | 0.4580 | 0.4675 | 0.4642 | 0.4601 | 0.0071 |
| enhancers | 0.4902 | 0.4993 | 0.5120 | 0.4986 | 0.5000 | 0.0086 |
| promoter no tata | 0.7577 | 0.7642 | 0.7543 | 0.7598 | 0.7590 | 0.0041 |
| promoter tata | 0.6855 | 0.6707 | 0.6552 | 0.7086 | 0.6800 | 0.0225 |
| promoter all | 0.7342 | 0.7396 | 0.7333 | 0.7566 | 0.7409 | 0.0108 |

Table G.3: Performance of DNAMotifTokenizer on Genomic Benchmarks, across 4 seeds

| Dataset | Seed 1 | Seed 2 | Seed 3 | Seed 4 | Mean | Std |
|---|---|---|---|---|---|---|
| demo human vs worm | 0.9213 | 0.9223 | 0.9221 | 0.9179 | 0.9209 | 0.0020 |
| dummy mouse enhancers ensembl | 0.4408 | 0.4533 | 0.5031 | 0.4683 | 0.4664 | 0.0257 |
| human enhancers ensembl | 0.7426 | 0.7371 | 0.7355 | 0.7371 | 0.7381 | 0.0032 |
| human nontata promoters | 0.7717 | 0.7781 | 0.7552 | 0.7573 | 0.7656 | 0.0109 |
| demo coding vs intergenomic seqs | 0.8248 | 0.8283 | 0.8244 | 0.8320 | 0.8274 | 0.0034 |
| drosophila enhancers stark | 0.3872 | 0.3899 | 0.3386 | 0.5248 | 0.4101 | 0.0787 |
| human enhancers cohn | 0.4505 | 0.4695 | 0.4579 | 0.4436 | 0.4554 | 0.0109 |
| human ensembl regulatory | 0.8629 | 0.8620 | 0.8624 | 0.8578 | 0.8613 | 0.0024 |
| human ocr ensembl | 0.5332 | 0.4986 | 0.5186 | 0.5069 | 0.5143 | 0.0147 |

Table G.4: Performance of DNAMotifTokenizer on SCREEN datasets, across 4 seeds

| Dataset | Seed 1 | Seed 2 | Seed 3 | Seed 4 | Mean | Std |
|---|---|---|---|---|---|---|
| CA-CTCF | 0.8822 | 0.8820 | 0.8782 | 0.8812 | 0.8809 | 0.0018 |
| pELS | 0.8961 | 0.8961 | 0.8981 | 0.8975 | 0.8970 | 0.0010 |
| CA | 0.8888 | 0.8843 | 0.8904 | 0.8900 | 0.8884 | 0.0027 |
| CA-H3K4me3 | 0.8795 | 0.8777 | 0.8764 | 0.8755 | 0.8773 | 0.0018 |
| CA-TF | 0.8380 | 0.8086 | 0.8368 | 0.8106 | 0.8235 | 0.0162 |
| TF | 0.8826 | 0.8847 | 0.8828 | 0.8831 | 0.8833 | 0.0009 |
| PLS | 0.9087 | 0.9037 | 0.9033 | 0.9018 | 0.9044 | 0.0031 |
| dELS | 0.9039 | 0.9037 | 0.9062 | 0.9045 | 0.9046 | 0.0012 |

Table G.5: Performance of DNAMotifTokenizer on DART-EVAL datasets, across 4 seeds

| Dataset | Seed 1 | Seed 2 | Seed 3 | Seed 4 | Mean | Std |
|---|---|---|---|---|---|---|
| task1 | 0.8647 | 0.8607 | 0.8625 | 0.8647 | 0.8632 | 0.0018 |
| task2 | 0.9571 | 0.9564 | 0.9560 | 0.9476 | 0.9543 | 0.0044 |
| GM12878 | 0.8206 | 0.8202 | 0.8077 | 0.8128 | 0.8153 | 0.0062 |
| H1ESC | 0.8202 | 0.8224 | 0.8163 | 0.8191 | 0.8195 | 0.0024 |
| HEPG2 | 0.8292 | 0.8292 | 0.8259 | 0.8292 | 0.8284 | 0.0017 |
| IMR90 | 0.7926 | 0.7934 | 0.7880 | 0.7802 | 0.7886 | 0.0058 |
| K562 | 0.8217 | 0.8104 | 0.8124 | 0.8182 | 0.8157 | 0.0050 |

# H METRICS

## H.1 MATTHEWS CORRELATION COEFFICIENT

The Matthews Correlation Coefficient (MCC) is a metric for evaluating binary classification performance, particularly useful for imbalanced datasets. It takes into account true positives (TP), true negatives (TN), false positives (FP), and false negatives (FN):

$$\text{MCC} = \frac{TP \cdot TN - FP \cdot FN}{\sqrt{(TP + FP)(TP + FN)(TN + FP)(TN + FN)}}.$$

MCC ranges from $-1$ to $+1$, where $+1$ indicates perfect prediction, $0$ corresponds to random guessing, and $-1$ indicates total disagreement between predictions and true labels. This makes MCC especially suitable for genomic classification tasks where class imbalance is common.

## H.2 JACCARD INDEX

The Jaccard similarity index (also known as Intersection over Union) is a measure of similarity between two sets. Given two sets $A$ and $B$, it is defined as:

$$J(A, B) = \frac{|A \cap B|}{|A \cup B|},$$

where $|A \cap B|$ is the number of elements common to both sets, and $|A \cup B|$ is the total number of elements in either set.

The Jaccard index ranges from 0 to 1, where 1 indicates identical sets and 0 indicates completely disjoint sets. This metric is widely used in genomics for comparing predicted regions with ground truth annotations, such as cCREs or TF binding sites.

## I  MODEL

The essential code and scripts are in the **supplementary material**.

The pre-trained model with DNAMotifTokenizer and example data are available on HuggingFace: https://huggingface.co/Anonymous-843q0u4q08/DNAMotifTokenizer

