# OpenReview forum: "DNAMotifTokenizer: Towards Biologically Informed Tokenization of Genomic Sequences"
_ICLR.cc/2026/Conference — Submitted to ICLR 2026_

### Official Review · Reviewer_zmZc · 2025-10-30

**Soundness:** 3
**Presentation:** 3
**Contribution:** 3
**Rating:** 6
**Confidence:** 3

**Summary:**

In this paper, the authors proposed DNAMotifTokenizer, a PWM-driven tokenizer that trims low-information flanks, handles reverse complements, and segments sequences via greedy trie matching. The tokenizer integrates seamlessly with BERT-style masked language modeling using motif-aware masking and an end-to-end, cache-friendly pretraining pipeline compatible with DNABERT. The goal is to replace purely k-mer tokenization with a more interpretable, biologically grounded alternative that preserves coordinate-level traceability. The given results suggest consistent gains on most of the existing benchmarks.

**Strengths:**

1. The author proposed a novel and interesting method to integrate the motif information to the tokenization, instead of eliminating such info as previous method did. It could bring more biologically useful information and help the language model investigate deeper in existing data.

2. The authors provide the implementation of the core idea, which aligns well with the paper.

3. The given results show that the proposed DNAMotifTokenizer can bring general improvement to most tasks.

4. Apart from normal content, the authors also took a deeper look into the existing BPE tokenizer for a further discussion and comparison with the proposed one, which improves the rationality.

**Weaknesses:**

1. Method: Heavy reliance on heuristics (length cap, flank-trimming thresholds, greedy matching) and PWM quality. Such components may bias learning and miss unknown motifs.

2. Experiment: Although the overall performance of the proposed method is very good, it shows obvious bad results on DART-EVAL. For example, in Table 4, the accuracy of the proposed DNAMotifTokenizer is much lower than the SOTA method. The authors are suggested to give the explanations of why this happens.

**Questions:**

How is the proposed DNAMotifTokenizer on more cross-species tasks? And what about more modern model architectures (e.g., llama, mamba), instead of BERTs only.

---

> ### Author Response · Authors · 2025-11-26
> **Reply to review by zmZc - Weakness 1**
>
> **Q1: Method: Heavy reliance on heuristics (length cap, flank-trimming thresholds, greedy matching) and PWM quality. Such components may bias learning and miss unknown motifs.**
>
> We appeciate the reviewer for pointing out this potential concern.
>
> The probability weight matrices (PWMs) we use are well curated from decades of wet-lab experiments, so we trust these collections and use them as the basis for our tokenization. We plot the motif length distribution of Jaspar Motif Library, and we noticed that the length typically range from 5 to 12. Therefore, we cap the motif length according to these observation, and kept motif length range from 5 to 12 nucleotides. This motif length cap also helps reduce vocabulary size, keeping it concise, informative, and expressive.
>
> For the probability threshold of 0.5, we follow the background probability reported in the PWM, which is globally 0.5 at each position. Regarding trimming wildcard ends: in PWM-based motif matching, the standard approach computes the dot product between a motif’s PWM and the one-hot encoded sequence. A sequence is considered a match if the resulting score exceeds a threshold. Positions at the ends without a dominant nucleotide (e.g., all probabilities = 0.25) are highly uncertain and essentially act as wildcards. We believe these positions have minimal impact, as they can represent any nucleotide. Including such wildcards would also exponentially increase the vocabulary size, so trimming them keeps the motif representations concise and efficient.
>
> Moreover, DNA sequence motifs are identified from wet-lab experiments, the uncertainty of nucleotide could be also introduced by technical bias and/or inconsistency from different experimental assay, that’s why we decided to focus on the nucleotides have higher confidence.
>
> We acknowledge that our algorithm is suboptimal in the **Conclusion** section from line 534 to 536. While current k-mer and BPE-based tokenizers offer flexibility, they struggle to represent the sparsity and uneven distribution of regulatory features [1]. Therefore, our work focuses on testing whether incorporating biologically meaningful priors, such as TF motifs, can effectively serve as predefined "words" in genomic tokenization. Our current goal is to explore the semantic structure of genomic sequences and understand how motifs can guide DNA language modeling. In our future work, we aim for a fully data-driven, automatic DNA tokenization algorithm.
>
> **Reference:**
>
> [1] Patel, Aman, et al. "DART-Eval: A comprehensive DNA language model evaluation benchmark on regulatory DNA." Advances in Neural Information Processing Systems 37 (2024): 62024-62061.

---

> ### Author Response · Authors · 2025-11-26
> **Reply to review by zmZc - Weakness 2**
>
> **Q2: Experiment: Although the overall performance of the proposed method is very good, it shows obvious bad results on DART-EVAL. For example, in Table 4, the accuracy of the proposed DNAMotifTokenizer is much lower than the SOTA method. The authors are suggested to give the explanations of why this happens.**
>
> We appeciate the reviewer for pointing out this potential consern!
>
> In DART-EVAL dataset, their task1 and task 2 datasets are both synthetic data, by inserting real cCRE sequences and one motif sequences into random background sequence. This modification makes each sequence in task1 and 2 naturally don’t have real motif distributed, combined with the potential noise introduced by random sequences, which may affect the classification performance for our method. We added corresponding explaination to our revised manuscript from line 469 to 470.
>
> We also did additional experiments to demonstrate the interpretability, generalizability, stability and complexity of our proposed DNAMotifTokenizer. We collected a new dataset from various brain cell types to showcase the interpretability of our model and tokenizer, which have been incorporated as a main **Figure 6** in **section 7.2** of our revised manuscript. We demonstrate the generalizability of our tokenizer in cross-species prediction tasks, which have been incorporated into the main text under **Section 7.2** and **Table F.6**. We used another three seeds to tokenize the DNA sequence and train models, which results in stable performance. These results have been incorporated into the main text in **Section 7.2** and **Table G.1~G.5**. We analyzed the computational complexity in **Section 7.3**.

---

> ### Author Response · Authors · 2025-11-27
> **Reply to review by zmZc - Questions**
>
> **How is the proposed DNAMotifTokenizer on more cross-species tasks? And what about more modern model architectures (e.g., llama, mamba), instead of BERTs only.**
>
> We thank the author for these questions!
> To further evaluate cross-species generalizability, we have added fine-tuning results on Yeast and Mouse datasets from the GUE benchmark. Compared with k-mer and BPE tokenizers trained on multiple species, DNAMotifTokenizer achieves comparable or superior performance on these cross-species prediction tasks. Specifically, DNAMotifTokenizer obtains an average MCC of 0.4662 on Yeast and 0.5509 on Mouse, as shown in the table below.
>
> | Model            | Epigenetic Marks Prediction (Yeast) | Transcription Factor Prediction (Mouse) |
> |------------------|--------------------------------------|------------------------------------------|
> | 3mer (stride=1)  | 0.4399                               | 0.4034                                   |
> | 6mer (stride=1)  | 0.4301                               | 0.4466                                   |
> | 6mer (stride=6)  | 0.3711                               | 0.3056                                   |
> | BPE (DNABERT-2)  | **0.4670**                           | **0.5509**                               |
> | our (longest)    | 0.4639                        | 0.5306                            |
> | our (shortest)   |0.4609                        | 0.5458                            |
> | our              | **0.4662**                           | **0.5509**                               |
>
> We have incorporated these new results into our revised manuscript under **Section 7.2** and **Table F.6**.
>
> Regarding the architectures: in this work, our primary focus is on the design and understanding of tokenizers for genomic sequences. This serves as a theoretical foundation for our future development of genomic language models. To ensure a fair and controlled comparison, all pretraining experiments were conducted under consistent and well-defined settings.
>
> Most existing genomic language models—such as DNABERT1/2 and Nucleotide Transformer—are based on the BERT architecture. Therefore, we chose BERT as a practical trade-off between computational cost and comparability with prior work. This allows us to attribute performance differences primarily to the tokenizer design rather than architectural changes.
>
> We appreciate the reviewer's suggestion regarding alternative architectures such as LLAMA and Mamba. We agree that exploring these architectures is valuable, and we plan to investigate them in future work to further evaluate scalability and generalizability across model families.

---

### Official Review · Reviewer_UhJ2 · 2025-10-31

**Soundness:** 1
**Presentation:** 2
**Contribution:** 1
**Rating:** 2
**Confidence:** 5

**Summary:**

This paper proposes DNAMotifTokenizer, a motif-aware tokenization scheme for genomic sequences that integrates curated TF-motif priors and cCRE annotations to build a vocabulary of motif tokens. The work evaluates across several public genomics benchmarks (e.g., Genomic Benchmarks, GUE, NT-benchmarks, DART-Eval).

**Strengths:**

1. Brings explicit biological priors (TF motifs, cRE) into the tokenization step, offering a more interpretable alternative to opaque subword units.
2. Provides multiple ablations on vocabulary size, segmentation strategies, and qualitative motif coverage

**Weaknesses:**

1. The paper’s own results indicate k-mer consistency better than BPE, and DNAMotifTokenizer’s average on NT-benchmarks remains notably below k-mer. This undercuts the central narrative that knowledge-injected tokenization improves fundamental understanding.

2. Converting PWMs via a fixed 0.5 threshold and trimming wildcard ends discards degenerate bases and positional uncertainty, which are biologically meaningful. This can fragment genuine motif families and reduce robustness to natural variation.

3. When multiple motifs overlap, the default random choice introduces non-determinism. The paper lacks multi-seed repeats and dispersion metrics to judge stability.

**Questions:**

Please see Weaknesses.

---

> ### Author Response · Authors · 2025-11-26
> **Reply to review by UhJ2 - Weakness 1**
>
> **Q1: The paper’s own results indicate k-mer consistency better than BPE, and DNAMotifTokenizer’s average on NT-benchmarks remains notably below k-mer. This undercuts the central narrative that knowledge-injected tokenization improves fundamental understanding.**
>
> We appreciate the reviewer for pointing out this potential issue. With the recent publication of various genomic models, several benchmarking datasets have been introduced; however, there is currently no consensus on a standard evaluation framework, nor has the robustness of these datasets been fully established. We acknowledge that our model's performance on the Nucleotide Transformer (NT) benchmarks differs from other datasets, yet we also observed that k-mer based tokenizers do not consistently outperform other methods across the broader landscape of benchmarks. Therefore, we argue that the efficacy of k-mer based tokenization cannot be comprehensively evaluated using the NT benchmarking dataset alone. Because of these concerns, we additionally performed finetuning on several other benchmark datasets, as reported. Across four out of five datasets, k-mer tokenization typically underperformed compared to the BPE family and our method.
>
> It has been demonstrated that TF motifs are highly conserved across 600 million years of bilaterian evolution [1]. The complexity of gene regulation often arises from the specific rearrangement and combination of these conserved motifs into context-specific regulatory elements, rather than the emergence of entirely new motifs [2]. While current k-mer and BPE-based tokenizers offer flexibility, they struggle to represent the sparsity and uneven distribution of regulatory features [3]. Therefore, our work focuses on testing whether incorporating biologically meaningful priors, such as TF motifs, can effectively serve as predefined "words" in genomic tokenization.
>
> We also did additional analyses to demonstrate the interpretability, generalizability, stability and complexity of our proposed DNAMotifTokenizer. We collected a new dataset from various brain cell types to showcase the interpretability of our model and tokenizer, which have been incorporated as a main **Figure 6** in **Section 7.2**. We demonstrate the generalizability of our tokenizer in cross-species prediction tasks, and we used another three seeds to tokenize the DNA sequence and train models with our DNAMotifTokenizer, which results in stable performance. These results have been incorporated into the main text in **Section 7.2**. We analyzed the computational complexity in **Section 7.3**.
>
> **Reference:**
>
> [1] Nitta, Kazuhiro R., et al. "Conservation of transcription factor binding specificities across 600 million years of bilateria evolution." elife 4 (2015): e04837.
>
> [2] Wong, Emily S., et al. "Deep conservation of the enhancer regulatory code in animals." Science 370.6517 (2020): eaax8137.
>
> [3] Patel, Aman, et al. "DART-Eval: A comprehensive DNA language model evaluation benchmark on regulatory DNA." Advances in Neural Information Processing Systems 37 (2024): 62024-62061.

---

> ### Author Response · Authors · 2025-11-26
> **Reply to review by UhJ2 - Weakness 2**
>
> **Q2: Converting PWMs via a fixed 0.5 threshold and trimming wildcard ends discards degenerate bases and positional uncertainty, which are biologically meaningful. This can fragment genuine motif families and reduce robustness to natural variation.**
>
> We appreciate the reviewer for  pointing out this potential concern! We carefully considered both the probability threshold and whether to trim wildcard ends while designing our algorithm.
>
> For the probability threshold 0.5, we defer to the background probability reported to the probability weight matrix, which is globally 0.5 at each position.
>
> For trimming wildcard ends, in PWM-based motif matching, people typically compute the dot product between the motif’s probability weight matrix and the one-hot encoded sequence. A sequence is considered a match if the result exceeds a threshold.
> Positions at the ends without a dominant nucleotide (e.g., all probabilities = 0.25) are highly uncertain and essentially act as wildcards. We believe these positions have minimal impact, since they can represent any nucleotide. Including such wildcards at the ends would also increase the vocabulary size exponentially, so we think keeping motifs concise helps maintain an efficient vocabulary.
>
> In addition, DNA sequence motifs are identified from wet-lab experiments, the uncertainty of nucleotide could be also introduced by technical bias and/or inconsistency from different experimental assay, that’s why we decided to focus on the nucleotides have higher confidence.

---

> ### Author Response · Authors · 2025-11-26
> **Reply to review by UhJ2 - Weakness 3 (Part 1)**
>
> **Q3: When multiple motifs overlap, the default random choice introduces non-determinism. The paper lacks multi-seed repeats and dispersion metrics to judge stability.**
>
> We thank the reviewer for pointing this out! We performed another 3 groups of pretraining for our DNAMotifTokenizer by setting another three different seeds, and finetuned on the same five Benchmark datasets. These results demonstrate the stability of our proposed method. We summarized our finetuning performance for our DNAMotifTokenizer with 4 different seeds into tables as below, and these results are also added to the **Section 7.2** and **Table G.1 ~ G.5** in our revised manuscript.
>
> **Table 1: Performance of DNAMotifTokenizer on GUE across 4 seeds**
>
> | **Dataset**        | **Seed 1** | **Seed 2** | **Seed 3** | **Seed 4** | **Mean** | **Std** |
> |-------------------|-----------|-----------|-----------|-----------|---------|---------|
> | prom_core_all      | 0.6488    | 0.6362    | 0.6484    | 0.6483    | 0.6454  | 0.0063 |
> | prom_core_notata   | 0.6564    | 0.6530    | 0.6496    | 0.6613    | 0.6551  | 0.0051 |
> | prom_core_tata     | 0.7195    | 0.7164    | 0.7164    | 0.6999    | 0.7131  | 0.0091 |
> | prom_300_all       | 0.8538    | 0.8412    | 0.8307    | 0.8351    | 0.8402  | 0.0095 |
> | prom_300_notata    | 0.9062    | 0.8967    | 0.9003    | 0.8994    | 0.9007  | 0.0040 |
> | prom_300_tata      | 0.6441    | 0.6049    | 0.6147    | 0.5625    | 0.6066  | 0.0332 |
> | reconstructed      | 0.7693    | 0.7620    | 0.7232    | 0.7474    | 0.7505  | 0.0202 |
> | tf_0               | 0.6406    | 0.6326    | 0.6631    | 0.6277    | 0.6410  | 0.0153 |
> | tf_1               | 0.6670    | 0.6985    | 0.7011    | 0.6871    | 0.6884  | 0.0152 |
> | tf_2               | 0.5179    | 0.5715    | 0.5037    | 0.5000    | 0.5233  | 0.0327 |
> | tf_3               | 0.4651    | 0.4742    | 0.4401    | 0.4431    | 0.4556  | 0.0162 |
> | tf_4               | 0.6888    | 0.7039    | 0.6725    | 0.6778    | 0.6858  | 0.0138 |
>
>
> **Table 2: Performance of DNAMotifTokenizer on Nucleotide Transformer Benchmarks across 4 seeds**
>
> | **Dataset**                 | **Seed 1** | **Seed 2** | **Seed 3** | **Seed 4** | **Mean** | **Std**  |
> |-----------------------------|-----------|-----------|-----------|-----------|---------|---------|
> | H2AFZ                        | 0.4835    | 0.5106    | 0.4831    | 0.4888    | 0.4915  | 0.0122  |
> | H3K27ac                      | 0.4892    | 0.4663    | 0.4579    | 0.4850    | 0.4746  | 0.0142  |
> | H3K27me3                     | 0.5994    | 0.6002    | 0.5970    | 0.6014    | 0.5995  | 0.0018  |
> | H3K36me3                     | 0.5915    | 0.5777    | 0.5852    | 0.5957    | 0.5875  | 0.0074  |
> | H3K4me1                      | 0.4814    | 0.4762    | 0.4763    | 0.4944    | 0.4821  | 0.0083  |
> | H3K4me2                      | 0.5775    | 0.5938    | 0.5771    | 0.5816    | 0.5825  | 0.0076  |
> | H3K4me3                      | 0.6701    | 0.6705    | 0.6612    | 0.6748    | 0.6692  | 0.0058  |
> | H3K9ac                       | 0.5434    | 0.5437    | 0.5426    | 0.5452    | 0.5437  | 0.0011  |
> | H3K9me3                      | 0.4170    | 0.4218    | 0.4202    | 0.4279    | 0.4217  | 0.0044  |
> | H4K20me1                     | 0.6324    | 0.6277    | 0.6179    | 0.6183    | 0.6241  | 0.0069  |
> | splice sites donors           | 0.7855    | 0.7881    | 0.7994    | 0.7304    | 0.7759  | 0.0306  |
> | splice sites acceptors        | 0.7309    | 0.7288    | 0.7653    | 0.7350    | 0.7400  | 0.0165  |
> | splice sites all              | 0.7119    | 0.7072    | 0.7598    | 0.6592    | 0.7095  | 0.0418  |
> | enhancers types               | 0.4507    | 0.4580    | 0.4675    | 0.4642    | 0.4601  | 0.0071  |
> | enhancers                     | 0.4902    | 0.4993    | 0.5120    | 0.4986    | 0.5000  | 0.0086  |
> | promoter no tata              | 0.7577    | 0.7642    | 0.7543    | 0.7598    | 0.7590  | 0.0041  |
> | promoter tata                 | 0.6855    | 0.6707    | 0.6552    | 0.7086    | 0.6800  | 0.0225  |
> | promoter all                  | 0.7342    | 0.7396    | 0.7333    | 0.7566    | 0.7409  | 0.0108  |

---

> ### Author Response · Authors · 2025-11-26
> **Reply to review by UhJ2 - Weakness 3 (Part 2)**
>
> **Table 3: Performance of DNAMotifTokenizer on Genomic Benchmarks across 4 seeds**
>
> | **Dataset**                        | **Seed 1** | **Seed 2** | **Seed 3** | **Seed 4** | **Mean** | **Std**  |
> |-----------------------------------|-----------|-----------|-----------|-----------|---------|---------|
> | demo human vs worm                  | 0.9213    | 0.9223    | 0.9221    | 0.9179    | 0.9209  | 0.0020  |
> | dummy mouse enhancers ensembl       | 0.4408    | 0.4533    | 0.5031    | 0.4683    | 0.4664  | 0.0257  |
> | human enhancers ensembl             | 0.7426    | 0.7371    | 0.7355    | 0.7371    | 0.7381  | 0.0032  |
> | human nontata promoters             | 0.7717    | 0.7781    | 0.7552    | 0.7573    | 0.7656  | 0.0109  |
> | demo coding vs intergenomic seqs    | 0.8248    | 0.8283    | 0.8244    | 0.8320    | 0.8274  | 0.0034  |
> | drosophila enhancers stark          | 0.3872    | 0.3899    | 0.3386    | 0.5248    | 0.4101  | 0.0787  |
> | human enhancers cohn                | 0.4505    | 0.4695    | 0.4579    | 0.4436    | 0.4554  | 0.0109  |
> | human ensembl regulatory            | 0.8629    | 0.8620    | 0.8624    | 0.8578    | 0.8613  | 0.0024  |
> | human ocr ensembl                   | 0.5332    | 0.4986    | 0.5186    | 0.5069    | 0.5143  | 0.0147  |
>
> **Table 4: Performance of DNAMotifTokenizer on SCREEN datasets across 4 seeds**
>
> | **Dataset**      | **Seed 1** | **Seed 2** | **Seed 3** | **Seed 4** | **Mean** | **Std**  |
> |-----------------|-----------|-----------|-----------|-----------|---------|---------|
> | CA-CTCF          | 0.8822    | 0.8820    | 0.8782    | 0.8812    | 0.8809  | 0.0018  |
> | pELS             | 0.8961    | 0.8961    | 0.8981    | 0.8975    | 0.8970  | 0.0010  |
> | CA               | 0.8888    | 0.8843    | 0.8904    | 0.8900    | 0.8884  | 0.0027  |
> | CA-H3K4me3       | 0.8795    | 0.8777    | 0.8764    | 0.8755    | 0.8773  | 0.0018  |
> | CA-TF            | 0.8380    | 0.8086    | 0.8368    | 0.8106    | 0.8235  | 0.0162  |
> | TF               | 0.8826    | 0.8847    | 0.8828    | 0.8831    | 0.8833  | 0.0009  |
> | PLS              | 0.9087    | 0.9037    | 0.9033    | 0.9018    | 0.9044  | 0.0031  |
> | dELS             | 0.9039    | 0.9037    | 0.9062    | 0.9045    | 0.9046  | 0.0012  |
>
> **Table 5: Performance of DNAMotifTokenizer on DART-EVAL datasets across 4 seeds**
>
> | **Dataset**  | **Seed 1** | **Seed 2** | **Seed 3** | **Seed 4** | **Mean** | **Std**  |
> |-------------|-----------|-----------|-----------|-----------|---------|---------|
> | task1       | 0.8647    | 0.8607    | 0.8625    | 0.8647    | 0.8632  | 0.0018  |
> | task2       | 0.9571    | 0.9564    | 0.9560    | 0.9476    | 0.9543  | 0.0044  |
> | GM12878     | 0.8206    | 0.8202    | 0.8077    | 0.8128    | 0.8153  | 0.0062  |
> | H1ESC       | 0.8202    | 0.8224    | 0.8163    | 0.8191    | 0.8195  | 0.0024  |
> | HEPG2       | 0.8292    | 0.8292    | 0.8259    | 0.8292    | 0.8284  | 0.0017  |
> | IMR90       | 0.7926    | 0.7934    | 0.7880    | 0.7802    | 0.7886  | 0.0058  |
> | K562        | 0.8217    | 0.8104    | 0.8124    | 0.8182    | 0.8157  | 0.0050  |

---

### Official Review · Reviewer_xLqN · 2025-11-01

**Soundness:** 2
**Presentation:** 1
**Contribution:** 2
**Rating:** 4
**Confidence:** 4

**Summary:**

The paper benchmarks DNA tokenization strategies under a controlled pretraining setup and proposes DNAMotifTokenizer, which injects motif knowledge into the vocabulary and uses a greedy, locally flexible matching procedure. The results suggest that larger BPE vocabularies are not necessarily better, and that training the tokenizer on biologically informative subsets (e.g., motifs or cCRE regions) can perform comparably to training on the entire genome.

**Strengths:**

1. The proposed tokenizer is conceptually simple and biologically informed, with clear pseudocode that enhances reproducibility.
2. The experimental design of this study is rigorous as it meticulously isolates the impact of tokenization by matching computational FLOPs, model architecture, and fine-tuning pipelines across all comparisons (GUE, SCREEN, DART-Eval, Genomic Benchmarks, NT Benchmarks).

**Weaknesses:**

1. The technical presentation of this work requires further efforts.  The paper presents both a benchmark and a new method in a 9-page paper. The direct result is that the benchmark is not comprehensive and the analysis of the method is not enough.
2. The experimental results show small absolute gains, and the variance is not reported, e.g. some improvements are ≤ 0.0005 in absolute terms.
3. The 0-2 bp offset and random tie-breaking are reasonable, but their stability and computational complexity are not fully characterized. More discussion are needed.
4. The figure captions are insufficiently detailed, lacking explanations for the individual subpanels (a, b, c, etc.), and the absence of descriptive legends hinders the interpretation of key elements.

**Questions:**

1. What's the algorithm's sensitivity to parameters like the 0-2 bp offset and the tie-breaking mechanism? The performance gains appear modest relative to the added complexity compared to standard BPE. Please discuss the specific scenarios where this complexity is justified.
·What is the primary intended contribution of this paper — a new method or a benchmark? The current structure does not fully align with either goal: as a benchmark paper, the experimental section of the main text is not enough; as a method paper, the narrative structure is not very reasonable.

---

> ### Author Response · Authors · 2025-11-26
> **Reply to review by xLqN - Weakness 1**
>
> We appreciate the reviewer’s careful reading, helpful comments, and constructive suggestions, which has significantly improved the presentation of our manuscript. We have carefully considered all comments from the reviewers and revised our manuscript accordingly.
>
> **Q1: The technical presentation of this work requires further efforts. The paper presents both a benchmark and a new method in a 9-page paper. The direct result is that the benchmark is not comprehensive and the analysis of the method is not enough.**
>
> We sincerely appreciate your valuable suggestion. We decided to move more benchmarking results to appendix, which allow us to provide more technical details of our proposed DNAMotifTokenier, and emphasize that biological prior knowledge could be used as predefined “words” for DNA tokenization.
>
> We did additional analyses to assess the generalizability, stability, complexity, and interpretability of our model and proposed tokenizer.
> - We collected a new dataset from various brain cell types to showcase the interpretability of our model and tokenizer, which have been incorporated as a main **Figure 6** in **Section 7.2** in revised manuscript.
> - We demonstrate the generalizability of our tokenizer in cross-species prediction tasks, which have been incorporated into the revised manuscript under **Section 7.2**, reported the results in **Table F.6**.
> - We used four different seeds to tokenize the DNA sequence and train models, which results in stable performance. These results have been incorporated into the revised manuscript under **Section 7.2** and **Table G.1 ~ G.5**.
> - We analyzed the computational complexity in revised manuscript in Section **7.3**.
>
> We hope these additional analyses and clarifications address the reviewer’s concerns and further enhance the rigor and completeness of our study. We are grateful for the reviewer’s thoughtful suggestions, which significantly improved the quality of this work.

---

> ### Author Response · Authors · 2025-11-26
> **Reply to review by xLqN - Weakness 2**
>
> **Q2: The experimental results show small absolute gains, and the variance is not reported, e.g. some improvements are ≤ 0.0005 in absolute terms.**
>
> We thank the reviewer for pointing this out. We have updated all the main and supplementary tables to include standard deviations (STDs). We also pretrained our models with another three different seeds, and noticed very small variance, demonstrating the stability of our methods. Those results have been included in our revised manuscript under **Section 7.2**, and **Table G.1 ~ G.5**. They have also been added to our response to Weakness 3.

---

> ### Author Response · Authors · 2025-11-26
> **Reply to review by xLqN - Weakness 3 (Part 1)**
>
> **Q3: The 0-2 bp offset and random tie-breaking are reasonable, but their stability and computational complexity are not fully characterized. More discussion are needed.**
>
> We thank the reviewer for pointing this out! We performed another three groups of pretraining for our DNAMotifTokenizer by setting another three different seeds, and finetuned on the same five Benchmark datasets. And we summarized our finetuning performance for our DNAMotifTokenizer with all four different seeds into tables as below, and these results are also added to the **Section 7.2** and Table **G.1 ~ G.5** in our revised manuscript.
>
>
> **Table 1: Performance of DNAMotifTokenizer on GUE across 4 seeds**
>
> | **Dataset**        | **Seed 1** | **Seed 2** | **Seed 3** | **Seed 4** | **Mean** | **Std** |
> |-------------------|-----------|-----------|-----------|-----------|---------|---------|
> | prom_core_all      | 0.6488    | 0.6362    | 0.6484    | 0.6483    | 0.6454  | 0.0063 |
> | prom_core_notata   | 0.6564    | 0.6530    | 0.6496    | 0.6613    | 0.6551  | 0.0051 |
> | prom_core_tata     | 0.7195    | 0.7164    | 0.7164    | 0.6999    | 0.7131  | 0.0091 |
> | prom_300_all       | 0.8538    | 0.8412    | 0.8307    | 0.8351    | 0.8402  | 0.0095 |
> | prom_300_notata    | 0.9062    | 0.8967    | 0.9003    | 0.8994    | 0.9007  | 0.0040 |
> | prom_300_tata      | 0.6441    | 0.6049    | 0.6147    | 0.5625    | 0.6066  | 0.0332 |
> | reconstructed      | 0.7693    | 0.7620    | 0.7232    | 0.7474    | 0.7505  | 0.0202 |
> | tf_0               | 0.6406    | 0.6326    | 0.6631    | 0.6277    | 0.6410  | 0.0153 |
> | tf_1               | 0.6670    | 0.6985    | 0.7011    | 0.6871    | 0.6884  | 0.0152 |
> | tf_2               | 0.5179    | 0.5715    | 0.5037    | 0.5000    | 0.5233  | 0.0327 |
> | tf_3               | 0.4651    | 0.4742    | 0.4401    | 0.4431    | 0.4556  | 0.0162 |
> | tf_4               | 0.6888    | 0.7039    | 0.6725    | 0.6778    | 0.6858  | 0.0138 |
>
>
> **Table 2: Performance of DNAMotifTokenizer on Nucleotide Transformer Benchmarks across 4 seeds**
>
> | **Dataset**                 | **Seed 1** | **Seed 2** | **Seed 3** | **Seed 4** | **Mean** | **Std**  |
> |-----------------------------|-----------|-----------|-----------|-----------|---------|---------|
> | H2AFZ                        | 0.4835    | 0.5106    | 0.4831    | 0.4888    | 0.4915  | 0.0122  |
> | H3K27ac                      | 0.4892    | 0.4663    | 0.4579    | 0.4850    | 0.4746  | 0.0142  |
> | H3K27me3                     | 0.5994    | 0.6002    | 0.5970    | 0.6014    | 0.5995  | 0.0018  |
> | H3K36me3                     | 0.5915    | 0.5777    | 0.5852    | 0.5957    | 0.5875  | 0.0074  |
> | H3K4me1                      | 0.4814    | 0.4762    | 0.4763    | 0.4944    | 0.4821  | 0.0083  |
> | H3K4me2                      | 0.5775    | 0.5938    | 0.5771    | 0.5816    | 0.5825  | 0.0076  |
> | H3K4me3                      | 0.6701    | 0.6705    | 0.6612    | 0.6748    | 0.6692  | 0.0058  |
> | H3K9ac                       | 0.5434    | 0.5437    | 0.5426    | 0.5452    | 0.5437  | 0.0011  |
> | H3K9me3                      | 0.4170    | 0.4218    | 0.4202    | 0.4279    | 0.4217  | 0.0044  |
> | H4K20me1                     | 0.6324    | 0.6277    | 0.6179    | 0.6183    | 0.6241  | 0.0069  |
> | splice sites donors           | 0.7855    | 0.7881    | 0.7994    | 0.7304    | 0.7759  | 0.0306  |
> | splice sites acceptors        | 0.7309    | 0.7288    | 0.7653    | 0.7350    | 0.7400  | 0.0165  |
> | splice sites all              | 0.7119    | 0.7072    | 0.7598    | 0.6592    | 0.7095  | 0.0418  |
> | enhancers types               | 0.4507    | 0.4580    | 0.4675    | 0.4642    | 0.4601  | 0.0071  |
> | enhancers                     | 0.4902    | 0.4993    | 0.5120    | 0.4986    | 0.5000  | 0.0086  |
> | promoter no tata              | 0.7577    | 0.7642    | 0.7543    | 0.7598    | 0.7590  | 0.0041  |
> | promoter tata                 | 0.6855    | 0.6707    | 0.6552    | 0.7086    | 0.6800  | 0.0225  |
> | promoter all                  | 0.7342    | 0.7396    | 0.7333    | 0.7566    | 0.7409  | 0.0108  |

---

> ### Author Response · Authors · 2025-11-26
> **Reply to review by xLqN - Weakness 3 (Part 2)**
>
> **Table 3: Performance of DNAMotifTokenizer on Genomic Benchmarks across 4 seeds**
>
> | **Dataset**                        | **Seed 1** | **Seed 2** | **Seed 3** | **Seed 4** | **Mean** | **Std**  |
> |-----------------------------------|-----------|-----------|-----------|-----------|---------|---------|
> | demo human vs worm                  | 0.9213    | 0.9223    | 0.9221    | 0.9179    | 0.9209  | 0.0020  |
> | dummy mouse enhancers ensembl       | 0.4408    | 0.4533    | 0.5031    | 0.4683    | 0.4664  | 0.0257  |
> | human enhancers ensembl             | 0.7426    | 0.7371    | 0.7355    | 0.7371    | 0.7381  | 0.0032  |
> | human nontata promoters             | 0.7717    | 0.7781    | 0.7552    | 0.7573    | 0.7656  | 0.0109  |
> | demo coding vs intergenomic seqs    | 0.8248    | 0.8283    | 0.8244    | 0.8320    | 0.8274  | 0.0034  |
> | drosophila enhancers stark          | 0.3872    | 0.3899    | 0.3386    | 0.5248    | 0.4101  | 0.0787  |
> | human enhancers cohn                | 0.4505    | 0.4695    | 0.4579    | 0.4436    | 0.4554  | 0.0109  |
> | human ensembl regulatory            | 0.8629    | 0.8620    | 0.8624    | 0.8578    | 0.8613  | 0.0024  |
> | human ocr ensembl                   | 0.5332    | 0.4986    | 0.5186    | 0.5069    | 0.5143  | 0.0147  |
>
> **Table 4: Performance of DNAMotifTokenizer on SCREEN datasets across 4 seeds**
>
> | **Dataset**      | **Seed 1** | **Seed 2** | **Seed 3** | **Seed 4** | **Mean** | **Std**  |
> |-----------------|-----------|-----------|-----------|-----------|---------|---------|
> | CA-CTCF          | 0.8822    | 0.8820    | 0.8782    | 0.8812    | 0.8809  | 0.0018  |
> | pELS             | 0.8961    | 0.8961    | 0.8981    | 0.8975    | 0.8970  | 0.0010  |
> | CA               | 0.8888    | 0.8843    | 0.8904    | 0.8900    | 0.8884  | 0.0027  |
> | CA-H3K4me3       | 0.8795    | 0.8777    | 0.8764    | 0.8755    | 0.8773  | 0.0018  |
> | CA-TF            | 0.8380    | 0.8086    | 0.8368    | 0.8106    | 0.8235  | 0.0162  |
> | TF               | 0.8826    | 0.8847    | 0.8828    | 0.8831    | 0.8833  | 0.0009  |
> | PLS              | 0.9087    | 0.9037    | 0.9033    | 0.9018    | 0.9044  | 0.0031  |
> | dELS             | 0.9039    | 0.9037    | 0.9062    | 0.9045    | 0.9046  | 0.0012  |
>
> **Table 5: Performance of DNAMotifTokenizer on DART-EVAL datasets across 4 seeds**
>
> | **Dataset**  | **Seed 1** | **Seed 2** | **Seed 3** | **Seed 4** | **Mean** | **Std**  |
> |-------------|-----------|-----------|-----------|-----------|---------|---------|
> | task1       | 0.8647    | 0.8607    | 0.8625    | 0.8647    | 0.8632  | 0.0018  |
> | task2       | 0.9571    | 0.9564    | 0.9560    | 0.9476    | 0.9543  | 0.0044  |
> | GM12878     | 0.8206    | 0.8202    | 0.8077    | 0.8128    | 0.8153  | 0.0062  |
> | H1ESC       | 0.8202    | 0.8224    | 0.8163    | 0.8191    | 0.8195  | 0.0024  |
> | HEPG2       | 0.8292    | 0.8292    | 0.8259    | 0.8292    | 0.8284  | 0.0017  |
> | IMR90       | 0.7926    | 0.7934    | 0.7880    | 0.7802    | 0.7886  | 0.0058  |
> | K562        | 0.8217    | 0.8104    | 0.8124    | 0.8182    | 0.8157  | 0.0050  |
>
> ---
>
> We also analyzed the time and space complexity as below, and it has been included into **Section 7.3** in our revised manuscript.
>
> **Time Complexity:**
> The average token length in our vocabulary is approximately 8.3 nucleotides.
> For a genomic sequence of length $n$, the tokenizer proceeds sequentially from left to right, resulting in approximately $O\left(\frac{n}{8.3}\right)$ tokenization steps. At each position, the tokenizer queries a Trie to identify motif tokens with lengths ranging from 4 up to $\text{Maxlen}$. If no motif is found, the tokenizer defaults to 3-mer or 1-mer tokens. The worst-case Trie lookup cost is $O(\text{Maxlen}^2)$.
> Considering potential shifts by 1 or 2 nucleotides at each position, the upper bound of the total time complexity is:
>
> $$
> O\left(\frac{n}{8.3} \cdot \text{Maxlen}^2\right)
> $$
> In our implementation, we set $\text{Maxlen} = 12$.
>
> ---
>
> **Space Complexity:**
> Let $V$ denote the vocabulary size and $L$ the average token length.
> The Trie storing all motif tokens requires $O(V \cdot L)$ space, while the lookup table mapping motifs to token IDs takes at most $O(V)$. During tokenization, the output tokens occupy $O\left(\frac{n}{8.3}\right)$ space.
>
> Therefore, the total space complexity is:
>
> $$
> O\left(\frac{n}{8.3} + V \cdot L\right)
> $$
> In our implementation:
> - $V = 901$
> - Average token length $L \approx 8.3$
> - Number of motif tokens stored in the Trie is 827

---

> ### Author Response · Authors · 2025-11-26
> **Reply to review by xLqN - Weakness 4 & Questions**
>
> **Q4: The figure captions are insufficiently detailed, lacking explanations for the individual subpanels (a, b, c, etc.), and the absence of descriptive legends hinders the interpretation of key elements.**
>
> Thank you for your careful review and pointing this out! We’ve adding clearer captions for each of the figure in our revised manuscript.
>
> ---
>
> **Questions: What's the algorithm's sensitivity to parameters like the 0-2 bp offset and the tie-breaking mechanism? The performance gains appear modest relative to the added complexity compared to standard BPE. Please discuss the specific scenarios where this complexity is justified. ·What is the primary intended contribution of this paper — a new method or a benchmark? The current structure does not fully align with either goal: as a benchmark paper, the experimental section of the main text is not enough; as a method paper, the narrative structure is not very reasonable.**
>
> We appreciate these constructive suggestions! We have improved our manuscript by incorporating more detailed assessment of our methods, including generalizability, stability, complexity, and interpretability.
>
> It has been demonstrated that TF motifs are highly conserved across 600 million years of bilaterian evolution [1]. The complexity of gene regulation often arises from the specific rearrangement and combination of these conserved motifs into context-specific regulatory elements, rather than the emergence of entirely new motifs [2]. While current k-mer and BPE-based tokenizers offer flexibility, they struggle to represent the sparsity and uneven distribution of regulatory features [3]. Therefore, our work focuses on testing whether incorporating biologically meaningful priors, such as TF motifs, can effectively serve as predefined "words" in genomic tokenization. We have modified this narrative in Background Section from our revised manuscript in line 128 - 135.
>
> **Reference:**
>
> [1] Nitta, Kazuhiro R., et al. "Conservation of transcription factor binding specificities across 600 million years of bilateria evolution." elife 4 (2015): e04837.
>
> [2] Wong, Emily S., et al. "Deep conservation of the enhancer regulatory code in animals." Science 370.6517 (2020): eaax8137.
>
> [3] Patel, Aman, et al. "DART-Eval: A comprehensive DNA language model evaluation benchmark on regulatory DNA." Advances in Neural Information Processing Systems 37 (2024): 62024-62061.

---

### Official Review · Reviewer_DHL6 · 2025-11-02

**Soundness:** 3
**Presentation:** 3
**Contribution:** 3
**Rating:** 4
**Confidence:** 4

**Summary:**

The paper proposes a DNA sequence tokenizer called DNAMotifTokenizer, which is based on transcription factor binding sites (TF motifs). By directly incorporating biological motifs into the vocabulary, it aims to enhance the interpretability and task performance of DNA language models. The authors conducted extensive experiments under multiple benchmarks (GUE, DART-Eval, NT) to systematically evaluate the impact of different tokenization strategies on model performance.

**Strengths:**

- The core contribution is the hard-coding of biological prior knowledge (transcription factor motifs, TF motifs) into the vocabulary as "tokens," demonstrating performance gains across multiple benchmarks through experiments.
- Focuses on the core issue of DNA language models—the impact of tokenization strategies—and conducts systematic and reproducible comparative experiments.
- The introduction of biological priors (motifs, cCREs) enhances interpretability, showing consistent gains across multiple tasks.
- The experimental design is rigorous, and the training budget and parameter scale are well-controlled, making the results reliable.

**Weaknesses:**

- The tokenization relies entirely on external databases (JASPAR, ENCODE), making the approach essentially "manual knowledge injection," which cannot adapt to unknown regions or new species.

- Limited Innovation: Using motifs as a vocabulary is an engineering improvement that is insufficient for a theoretical breakthrough. There is inadequate biological interpretative analysis, as the paper does not quantify the impact of motif tokens on the model's internal representations.

- About Generalizability: The use of a traditional BERT architecture and short sequence inputs restricts the model's generalizability.

Improvement Suggestions:
- The core contribution is the hard-coding of biological prior knowledge (transcription factor motifs, TF motifs) into the vocabulary as "tokens," demonstrating performance gains across multiple benchmarks through experiments.
- Focuses on the core issue of DNA language models—the impact of tokenization strategies—and conducts systematic and reproducible comparative experiments.
- The introduction of biological priors (motifs, cCREs) enhances interpretability, showing consistent gains across multiple tasks.
- The experimental design is rigorous, and the training budget and parameter scale are well-controlled, making the results reliable.
- Introduce a learnable motif discovery module to enable the model to have adaptive tokenization capabilities.
- Test generalizability on unknown regions or artificially mutated data.
- Provide interpretability metrics such as motif recovery rates and functional region enrichment.
- Explore the scalability of the method in long sequence modeling or generation tasks.

**Questions:**

please refer to the weaknesses part

---

> ### Author Response · Authors · 2025-11-26
> **Reply to review by DHL6 - Weakness 1**
>
> We appreciate the reviewer’s careful reading, helpful comments, and constructive suggestions, which has significantly improved the presentation of our manuscript.
>
> **Q1: The tokenization relies entirely on external databases (JASPAR, ENCODE), making the approach essentially "manual knowledge injection," which cannot adapt to unknown regions or new species.**
>
> We appreciate the reviewer for pointing out this potential limitation.
> We acknowledge that relying on external databases (e.g., JASPAR, ENCODE) may limit adaptability to unknown genomic regions or newly sequenced species. However, we regard these libraries as valuable sources of prior knowledge, curated from decades of rigorous research that represent our most current understanding of genomic regulation.
>
> It has been demonstrated that Transcription Factor (TF) motifs are highly conserved, not only among mammals but across 600 million years of bilaterian evolution [1]. The complexity of gene regulation often arises from the specific rearrangement and combination of these conserved motifs into context-specific regulatory elements, rather than the emergence of entirely new motifs [2]. While current k-mer and BPE-based tokenizers offer flexibility, they struggle to represent the sparsity and uneven distribution of regulatory features [3]. Therefore, our work focuses on testing whether incorporating biologically meaningful priors, such as TF motifs, can effectively serve as predefined "words" in genomic tokenization. Building on our positive results, we are conducting parallel ongoing projects to introduce more flexibility and scalability into DNA tokenization process, such as incorporating operators to represent potential mutations, developing data-driven and learnable motif tokenization modules.
>
> Furthermore, to empirically demonstrate the generalizability of our approach, we have included additional fine-tuning results on Yeast and Mouse datasets from the GUE dataset. Compared to k-mer and BPE tokenizers learnt from multiple species, DNAMotifTokenizer demonstrates comparable or superior performance in these cross-species predictions. Specifically, DNAMotifTokenizer achieves average MCC at 0.4662 in yeast, 0.5509 in mouse as shown in the table below. These additional results confirm the robustness and generalizability of DNAMotifTokenizer even outside human datasets.
>
> | Model            | Epigenetic Marks Prediction (Yeast) | Transcription Factor Prediction (Mouse) |
> |------------------|--------------------------------------|------------------------------------------|
> | 3mer (stride=1)  | 0.4399                               | 0.4034                                   |
> | 6mer (stride=1)  | 0.4301                               | 0.4466                                   |
> | 6mer (stride=6)  | 0.3711                               | 0.3056                                   |
> | BPE (DNABERT-2)  | **0.4670**                           | **0.5509**                               |
> | our (longest)    | 0.4639                        | 0.5306                            |
> | our (shortest)   |0.4609                        | 0.5458                            |
> | our              | **0.4662**                           | **0.5509**                               |
>
>
> We have incorporated these new results into our revised manuscript under **Section 7.2** and **Table F.6**.
>
> **References**
>
> [1] Nitta, Kazuhiro R., et al. "Conservation of transcription factor binding specificities across 600 million years of bilateria evolution." elife 4 (2015): e04837.
>
> [2] Wong, Emily S., et al. "Deep conservation of the enhancer regulatory code in animals." Science 370.6517 (2020): eaax8137.
>
> [3] Patel, Aman, et al. "DART-Eval: A comprehensive DNA language model evaluation benchmark on regulatory DNA." Advances in Neural Information Processing Systems 37 (2024): 62024-62061.

---

> ### Author Response · Authors · 2025-11-26
> **Reply to review by DHL6 - Weakness 2**
>
> **Q2: Limited Innovation: Using motifs as a vocabulary is an engineering improvement that is insufficient for a theoretical breakthrough. There is inadequate biological interpretative analysis, as the paper does not quantify the impact of motif tokens on the model's internal representations.**
>
> We appreciate this constructive suggestion!
> We collected single-nucleus ATAC-seq (snATAC-seq) data generated from three diverse brain cell types: intratelencephalic neurons from cortical layer 2/3 (ITL23), VIP-positive GABAergic neurons (VIP), and Microglia (MGC) [1]. We identified cell-type-specific ATAC-seq peak regions, which are served as prediction targets. To control the vocabulary size and achieve fair comparison, we then fine-tuned our pretrained DNAMotifTokenizer model (vocab size = 901) on this dataset.
>
> We have incorporated these results into the revised manuscript, including in **Figure 6** and **Section 7.2**.
>
> After fine-tuning, we performed token attribution analysis using Integrated Gradients [2] on the test set (10%). For each token, we computed its attribution score toward the true class label and then averaged the scores; Tokens with attribution scores farther away from zero—either positive or negative—indicate stronger influence on the model’s prediction. We ranked the average attribution scores for both k-mer and motif tokens, and we found that in our DNAMotifTokenizer model, most of the highly influential tokens correspond to motif tokens (in red), whereas the k-mer tokens (in blue) generally had attribution scores close to zero as shown in **Fig 6(b)**.
>
> We selected the top 200 most contributive motif tokens for each cell type.
> The Venn plot in **Figure 6(c)** indicates that DNAMotifTokenizer not only utilizes 143 motif tokens shared between three cell types, but also uses 25, 26, 34 cell-type-specific motif tokens for prediction in ITL23, VIP, MGC, respectively.
>
> Using only the test datasets, we then compared these most contributive motif tokens with the enriched motifs for each cell type reported in the original paper [1].  We showed the enrichment of matched motifs from three cell types as a heatmap in **Figure 6(d)**. From this analysis, we found that each cell type successfully captured its biologically enriched motifs. In ITL23 cell type, motifs from the WT1, RORa, Egr, and KLF families are captured and enriched, consistent with previous reports, whereas only WT1, Egr2, PGR, and KLF14 are captured in the VIP cell type. In MGC cell type, MAF(bZIP), ZNF(Zf) and Sox family motifs are captured and enriched, consistent with previous reports. These results demonstrate the interpretability of motif tokens introduced by DNAMotifTokenizer.
>
> These results further validate that our DNAMotifTokenizer is able to capture biologically meaningful elements for downstream tasks in a global manner.
>
> **References**
>
> [1] Li, Yang Eric, et al. "A comparative atlas of single-cell chromatin accessibility in the human brain." Science 382.6667 (2023): eadf7044.
>
> [2] Sundararajan, Mukund, Ankur Taly, and Qiqi Yan. "Axiomatic attribution for deep networks." International conference on machine learning. PMLR, 2017.

---

> ### Author Response · Authors · 2025-11-26
> **Reply to review by DHL6 - Weakness 3**
>
> **Q3: About Generalizability: The use of a traditional BERT architecture and short sequence inputs restricts the model's generalizability.**
>
> We thank the reviewer for pointing this out! We also considered the generalizability of our model. In this work, our primary focus is on the design and understanding of tokenizers for genomic sequences, which will serve as a theoretical foundation for future work on genomic language models. In this work, to ensure fair comparison, all pretraining experiments were conducted under well-controlled and consistent settings.
>
> Most existing genomic language models—such as DNABERT1 & 2 and Nucleotide Transformer—are based on the BERT architecture. Among these, DNABERT2 supports the longest input length, reaching ~3000 bp. We chose a similar architecture as a trade-off between computational resources, model size, and sequence length.
>
> In our future work, we plan to explore alternative architectures as well to further evaluate and improve the model’s generalizability.

---

> ### Author Response · Authors · 2025-11-26
> **Reply to review by DHL6 - Improvement Suggestions**
>
> We appreciate the reviewer’s kind words about our novelty in introducing biological prior knowledge, consistent gains across multiple tasks, and rigorous experimental design. To response to reviewer’s constructive suggestions, we did additional analyses to assess the generalizability, stability, complexity, and interpretability of our model.
>
> - We collected a new dataset from various brain cell types to showcase the interpretability of our model and tokenizer, which have been incorporated as a main **Figure 6** in **Section 7.2** in revised manuscript.
> - We demonstrate the generalizability of our tokenizer in cross-species prediction tasks, which have been incorporated into the revised manuscript under **Section 7.2**, reported the results in **Table F.6**.
> - We used four different seeds to tokenize the DNA sequence and train models, which results in stable performance. These results have been incorporated into the revised manuscript under **Section 7.2** and **Table G.1 ~ G.5**.
> - We analyzed the computational complexity in revised manuscript in Section **7.3**.
>
> We hope that these additional analyses and clarifications address the reviewer’s concerns and further strengthen our work. We sincerely appreciate the reviewer’s insightful feedback and the opportunity to improve our manuscript.

---

### Author Response · Authors · 2025-11-12
**No review comments received**

Dear review committee,

We have not received any review comments. Could you please provide guidance on how we should proceed?
Thanks a lot!

---

### Author Response · Authors · 2025-12-03
**Summary for the Area Chair’s Evaluation**

Dear Area Chair,

In this work, we did rigorous benchmarking and found that tokenizer choice induces task-specific trade-offs for DNA language models. We introduced **DNAMotifTokenizer**, which incorporates domain knowledge of DNA sequence motifs into the tokenization process. DNAMotifTokenizer consistently outperforms other tokenizers across diverse benchmarks, demonstrating that knowledge-infused tokenization is crucial for learning powerful, interpretable, and generalizable genomic representations.

We greatly appreciate reviewers' constructive comments and would like to provide a summary for your reference. All reviewers praised the core concept of integrating biological meaningful DNA sequence motifs directly into the tokenization vocabulary. They viewed this as a significant improvement over "opaque" subword units (BPE), offering better interpretability and preserving biological information. There is strong consensus (R1, R2) that the experimental design is scientifically sound. Reviewers (R1, R2, R3) appreciated the meticulous isolation of the tokenization variable by controlling for computational budget (FLOPs), model architecture, fine-tuning pipelines, and multiple ablations. The inclusion of clear pseudocode and implementation details was highlighted as a positive factor for reproducibility (R2, R4). Reviewers (R1, R4) also acknowledged that the method demonstrates performance improvements across multiple benchmarks and tasks.

In our rebuttal, we carefully addressed all the major concerns, and substantially strengthened the manuscript with additional experiments:

- ### **Generalizability to Unknown and New species:**

It has been demonstrated that Transcription Factor (TF) motifs are highly conserved, not only among mammals but across 600 million years of bilaterian evolution. While current k-mer and BPE-based tokenizers offer flexibility, they struggle to represent the sparsity and uneven distribution of DNA regulatory features. Therefore, our work focuses on testing whether incorporating biologically meaningful priors, such as TF motifs, can effectively serve as predefined "words" in genomic tokenization. We added fine-tuning experiments on Yeast and Mouse datasets, where DNAMotifTokenizer achieved comparable or superior performance to multi-species k-mer and BPE tokenizers, incorporated into the main text under **Section 7.2** and **Table F.6**.

Reliance on External DNA Motif Databases: we clarified that external databases such as JASPAR and ENCODE represent decades of curated biological knowledge and provide valuable priors. We also explained key design decisions in motif extraction, including trimming uncertain wildcard positions and picking probability thresholds, to reduce potential technical bias and prevent vocabulary inflation.

- ### **Stability and Complexity Estimation:**

We included multiple additional pretraining seeds to confirm stability which has been incorporated into the main text in **Section 7.2** and **Table G.1~G.5**. We provided a detailed complexity analysis in **Section 7.3** to clarify computational efficiency.


- ### **Lower Performance on Specific Task:**

Reviewer 4 pointed out a specific underperformance on the DART-EVAL benchmark compared to SOTA. We explained this result may be caused by potential bias from synthetic benchmarking data. By inserting motif sequences into random DNA sequences, which may not naturally represent the DNA motif sequence distribution in real genome.

- ### **Highlight Model Interpretability:**

Using snATAC-seq data from three brain cell types, we performed Integrated Gradients attribution and showed that the most influential tokens correspond to biologically enriched motifs reported in prior literature, highlighting improved interpretability, summarized as a main **Figure 6** in **Section 7.2**.

- ### **Future Directions:**

We appreciate constructive comments from reviewers. To ensure fair architectural comparison, we used a BERT-style encoder consistent with DNABERT1/2 and Nucleotide Transformer. In future work, we would like to extend to more modern model architectures. Building on our positive results, we are conducting parallel ongoing projects to introduce more flexibility and scalability into DNA tokenization process, such as incorporating operators to represent potential mutations, developing data-driven and learnable motif tokenization modules.

Altogether, this revised manuscript has clearly demonstrated introducing biological domain knowledge into the tokenization process improves both model performance and interpretation. Our proposed DNAMotifTokenizer is a robust, generalizable methods, offering insights that we believe will serve as a foundation for the development of future genomic language models.

---

### Meta-Review · Area_Chair_vNm1 · 2026-01-06

**Summary:**

This submission studies DNA tokenization as a first-class design choice for genomic language models. It first benchmarks k-mer and BPE tokenizers under tightly controlled pretraining (matched FLOPs, architecture, fine-tuning pipelines) across five benchmark suites, and then proposes DNAMotifTokenizer, which injects curated TF-motif knowledge into the vocabulary and performs greedy trie-based segmentation with small positional offsets. Reviewers broadly agree the question is important and that controlling for compute/architecture makes the comparison credible. The proposed motif-informed vocabulary is viewed as a meaningful step toward interpretability and biological traceability.

However, there is disagreement on whether the core novelty rises above “manual knowledge injection plus heuristics,” and whether the reported gains are consistently meaningful given variance, task-specific regressions (notably on parts of NT-benchmarks and DART-Eval relative to some baselines/SOTA), and reliance on curated motif databases.

**Reviewer Concerns:**

Addressed or partially addressed by the rebuttal

- Stability / variance (xLqN, UhJ2): Added multi-seed runs and reported stds, reducing concern about randomness and tiny gains.

- Interpretability (DHL6): Added IG attribution on snATAC-seq showing influential tokens align with known motifs.

Outstanding / not convincingly addressed

- Manual knowledge injection + database dependence (DHL6, UhJ2): Still no adaptive / data-driven motif discovery or clear evidence for missing/novel/mutated motifs.

- Biological fidelity of PWM discretization/trimming (UhJ2): Justified heuristics, but did not test robustness to degeneracy/mutations or motif-family fragmentation.

- Novelty (DHL6): Remains largely an engineering/benchmarking contribution without a principled explanation for why motif tokens win.

- Small effect sizes vs added complexity (xLqN): Hygiene improved, but still unclear when the extra machinery is worth it.

- NT-benchmarks tension (UhJ2): Broader benchmark argument doesn’t fully explain why k-mer wins there and what dataset properties drive reversals.

- Architecture scope (DHL6, zmZc): Still untested on longer contexts / modern architectures; acceptable, but limits impact claims.

**Reviewer Scores:**

DHL6: 4 (below threshold)
xLqN: 4 (below threshold)
UhJ2: 2 (reject)
zmZc: 6 (weak accept)

Despite meaningful rebuttal improvements (variance, seeds, interpretability add-on), the core concerns about non-adaptive prior injection, heuristic biological simplifications, and unclear practical significance vs complexity remain, with two reviewers still below threshold and one strongly negative.

---

### Decision · Program_Chairs · 2026-01-26

Reject